# TOWARDS IMPROVED SENTENCE REPRESENTATIONS USING TOKEN GRAPHS

**Krishna Sri Ipsit Mantri[1,2],\* Carola-Bibiane Schönlieb[3], Zorah Lähner[1,2], Moshe Eliasof[3,4]**
[1]University of Bonn
[2]Lamarr Institute for Machine Learning and Artificial Intelligence
[3]University of Cambridge
[4]Ben-Gurion University of the Negev
{kmantri, laehner}@uni-bonn.de, {cbs31, me532}@cam.ac.uk

## ABSTRACT

Obtaining a single-vector representation from a Large Language Model's (LLM) token-level outputs is a critical step for nearly all sentence-level tasks. However, standard pooling methods like mean or max aggregation treat tokens as an independent set, discarding the rich relational structure captured by the model's self-attention layers and making them susceptible to signal dilution. To address this, we introduce GLOT, a lightweight, structure-aware pooling module that reframes pooling as relational learning followed by aggregation. Operating on the outputs of a frozen LLM, GLOT first constructs a latent token-similarity graph, then refines token representations with a graph neural network, and finally aggregates them using a readout layer. Experimentally, our approach is remarkably robust and efficient: on a diagnostic stress test where 90% of tokens are random distractors, GLOT maintains over 97% accuracy while baseline methods collapse. Furthermore, it is competitive with state-of-the-art techniques on benchmarks like GLUE and MTEB with *20x fewer trainable parameters* and speeds up the training time by over *100x* compared with parameter-efficient fine-tuning methods. Supported by a theoretical analysis of its expressive power, our work shows that learning over token graphs is a powerful paradigm for the efficient adaptation of frozen LLMs. Our code is published at https://github.com/ipsitmantri/GLOT.

## 1 INTRODUCTION

Large Language Models (LLMs) (Raffel et al., 2020; Lewis et al., 2020; Brown et al., 2020; Touvron et al., 2023; Jiang et al., 2023) produce a sequence of token-level hidden states, yet many downstream applications require a single vector embedding to represent an entire sentence or document. Therefore, the process by which a sentence and its tokens' hidden states are collapsed into a single vector representation is critical. Standard practices typically rely on simple heuristics such as *mean*, *max*, or using a dedicated *[CLS]* token. While these pre-defined approaches have been dominant in the literature (Devlin et al., 2019; Liu et al., 2019; Reimers & Gurevych, 2019; Gao et al., 2021; Arora et al., 2017; Wang et al., 2024), they can also be vulnerable when only a small subset of tokens carries task-relevant signal amid many distractors, as has been recently studied in Brothers (2025).

Although Transformers (Vaswani et al., 2017) inherently model token interactions through self-attention, standard sentence-level representation techniques discard this rich relational structure, treating hidden states as an independent set of vectors. Indeed, this shortcoming was recently studied for Vision-Transformers (Dosovitskiy et al., 2021) in Brothers (2025), who proposed to learn a multilayer-perceptron (MLP)-based token scoring function. However, while this approach may correctly up-weight the word "good", it may fail to capture the effect of its negation with the word "not". This challenge is particularly acute for *decoder-only LMs* (e.g., GPT (Radford et al., 2019; Brown et al., 2020) or LLaMA (Touvron et al., 2023)), whose causal attention mechanism optimizes hidden states for next-token prediction rather than holistic sentence representation (Radford et al., 2019; Brown et al., 2020).

---

\*Work done while interning at Ben-Gurion University of the Negev

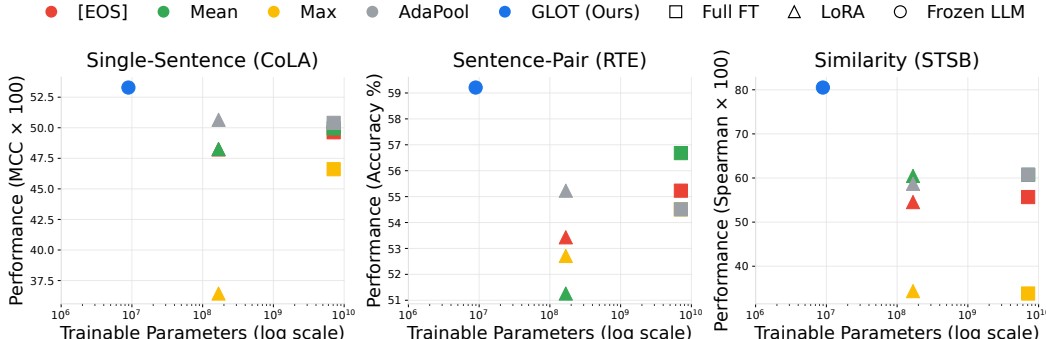

Figure 1: Fine-tuning large language models for sentence embeddings is computationally expensive. Our pooling method, GLOT, constructs a latent token-similarity graph from the outputs of a frozen model. It then refines token representations with a graph neural network before aggregation. This technique enables decoder-only models (like Mistral-7B), typically optimized for next-token prediction, to produce powerful sentence-level representations without requiring any fine-tuning.

Prior work shows that LLM token vectors have a strong directional bias: many of them point in similar directions, and seemingly unrelated words have embeddings with high similarity (Ethayarajh, 2019; Li et al., 2020). Therefore, sentence-level representations built on isolated tokens may be unreliable for semantic understanding tasks. While these shortcomings can be addressed by fine-tuning the entire model on downstream tasks, this approach is often computationally prohibitive for billion-parameter models (Lee et al., 2025; Gao et al., 2021). The immense cost of training, compounded by the need for extensive hyperparameter optimization, makes full fine-tuning impractical for many applications.

To bridge this gap, we reframe the problem of collapsing token hidden states into a sentence-level representation as *learning over token graphs*. To this end, we propose GLOT, a lightweight, structure-aware module that operates on the token hidden states produced by LLMs to obtain a sentence-level representation. Specifically, as illustrated in Figure 2, GLOT does the following: (i) constructs a token-similarity graph from the LLM hidden states, (ii) propagates information across the graph using a graph neural network, and (iii) aggregates the refined token representations using a readout mechanism. The LLM backbone remains entirely frozen; only the GNN module and a task-specific head are trained. This lightweight approach maintains a remarkably small memory footprint while equipping decoder-only LMs to perform as powerful text embedding models.

**Contributions.** Our contributions are as follows:

- We introduce a new conceptualization of sentence-level representation from LLM hidden states; rather than framing it as direct information compression like existing techniques, we envision a relational learning approach via GNNs. In addition, our framework generalizes common pooling methods like mean, max, and [CLS] pooling.

- We present GLOT, a compact and parameter-efficient module that enhances the performance of both encoder- and decoder-only frozen backbones with 20x fewer trainable parameters and over 100x faster training time than LLM fine-tuning-based methods.

- We provide extensive empirical validation for GLOT. Our evaluation shows that GLOT consistently outperforms pre-defined pooling and learning-based methods, across a wide range of tasks, including the **GLUE** benchmark for language understanding (Wang et al., 2018), long-text classification on **IMDB** (Maas et al., 2011), and seven diverse tasks from the large-scale **MTEB** benchmark (Muennighoff et al., 2023). Crucially, we introduce a novel diagnostic stress test that confirms GLOT's superior robustness to signal dilution, a key failure mode for other methods.

- We provide a detailed analysis of our method's components, demonstrating how the graph construction impacts performance and quantifying its substantial computational efficiency over fine-tuning methods.

## 2 RELATED WORK

**The Compressive Paradigm of Sentence-Level Representation.** To obtain sentence-level representations from LLMs, existing methods consider a compression problem: collapsing tokens' hidden states into a single vector. This paradigm usually encompasses pre-defined rules like *mean* or *max* selection, as well as learnable variants that learn token weights (Reimers & Gurevych, 2019; Gao et al., 2021; Xing et al., 2024; Lee et al., 2025; Brothers, 2025). While effective in some cases, these methods fundamentally discard relational structure. This can be derived from the theory of permutation-invariant functions on sets, as done in DeepSets (Zaheer et al., 2017), however, only looking at the tokens as completely independent items in a set does not paint the whole picture. As a result, these approaches implicitly assume the LLM has already embedded all necessary relational information. This assumption is often violated, especially for decoder-only models, which are optimized for next-token prediction rather than holistic sentence understanding (Radford et al., 2019; Brown et al., 2020). Indeed, recent work by Brothers (2025) shows such methods fail precisely because they compress *before* performing relational learning. Our work, GLOT, directly addresses this shortcoming by using advances from graph neural networks, which are also permutation invariant but can also encode relational information.

**Graph-Based Representations in NLP.** Graph Neural Networks (GNNs) are natural tools for relational learning; however, their prior applications in NLP differ from our problem of obtaining sentence-level representation using a frozen LLM. Many of these works use graphs to represent corpus-level tasks and solve them using GNNs rather than producing sentence-level embeddings. For example, Yao et al. (2019) builds a single word-occurrence-based graph over the corpus for text classification, and Huang et al. (2019) extends this approach for online inference and reduced memory consumption. Recent works propose the usage of attention and diffusion dynamics (Liu et al., 2021) and interleaving GNN and Transformer layers for improved text classification performance. Other approaches differ in their architecture or output format. Late-interaction models like ColBERT (Khattab & Zaharia, 2020) preserve token granularity but produce multi-vector representations incompatible with standard embedding interfaces. In contrast, GLOT is the first approach to construct a *latent token-similarity graph* directly from frozen LLM hidden states, and perform explicit relational learning *within the pooling head* to produce a single, robust sentence vector.

**Global Representations in Other Domains.** The challenge of creating a single, global representation from a set of features is not unique to NLP. In computer vision, pooling has long been a central component in convolutional neural networks (CNNs). While operations like max and average pooling are used throughout these models (Krizhevsky et al., 2012; He et al., 2016), global pooling is critical for producing a hoslistic representation. Techniques like global average pooling are used to collapse the final spatial feature maps into a single feature vector for classification, effectively summarizing the most salient features present in an image (Lin et al., 2013). In NLP, by contrast, pooling is often treated as a final, routine step. Our work, GLOT, challenges this view by demonstrating that a graph-neural-based sentence-level learning approach can unlock significant performance gains from frozen language models, opening a new direction for efficient sentence-level model adaptation.

**Positioning GLOT Relative to Prior Works.** Our work distinguishes itself from two primary streams of literature: learnable pooling and graph-based NLP. (a) *Relation to Learnable Pooling.* Recent learnable pooling methods, such as AdaPool (Brothers, 2025), operate fundamentally under a "DeepSets" paradigm (Zaheer et al., 2017). These approaches treat the token sequence as an independent set to be compressed. While effective for some tasks, this independence assumption fails to capture inter-token dependencies which are critical for resolving the signal dilution inherent in frozen LLM outputs. GLOT challenges this assumption by reframing pooling as *relational learning*. By explicitly modeling pairwise interactions via a GNN before aggregation, GLOT recovers structural dependencies that strictly independent pooling methods discard. (b) *Relation to Graph-Based NLP.* Unlike prior graph-based text encoding methods (e.g., TextGCN (Yao et al., 2019)), which typically rely on global corpus-level statistics or fixed syntactic dependency trees, GLOT introduces a *dynamic, latent graph construction* mechanism. GLOT builds semantic graphs on-the-fly based entirely on the intrinsic geometry of the frozen LLM's hidden states. This allows the model to recover rich, context-specific structural information without the computational overhead of external parsers or the rigidity of fixed syntactic trees.

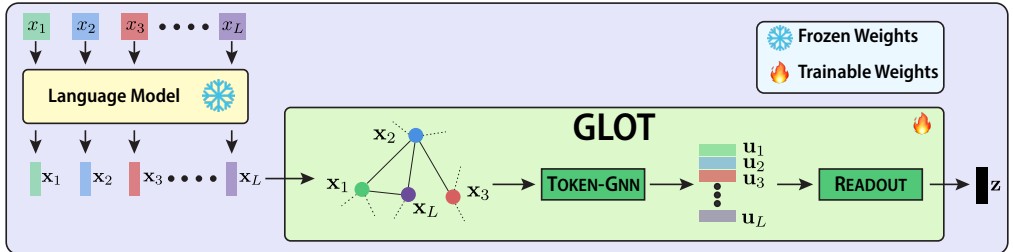

Figure 2: An overview of the GLOT pooling architecture. Given token hidden states from a frozen language model, our trainable module performs three stages : (1) it constructs a latent token-similarity graph, (2) a TOKEN-GNN performs relational learning to refine token representations, and (3) a readout layer aggregates the refined vectors into a final sentence representation, $\mathbf{z}$

## 3 METHOD

In this section we formalize and discuss the properties of our method. We start by providing essential notations and problem formulation in Section 3.1, followed by Section 3.2 where we present GLOT.

### 3.1 PROBLEM SETUP

Given a sequence of input tokens $[x_1, x_2, \cdots, x_L]$ and a frozen LLM, the task is to design a function $f_{\text{pool}}$, that maps the sequence of token-level hidden states $\mathbf{X} = [\mathbf{x}_1, \mathbf{x}_2, \cdots, \mathbf{x}_L] \in \mathbb{R}^{L \times d}$, to a single, sentence-level representation, $\mathbf{z} \in \mathbb{R}^D$. This vector $\mathbf{z}$ is a critical input for many downstream applications considered in this work, as follows:

- **Single-Sentence Classification.** For tasks like sentiment analysis, the vector $\mathbf{z}$ is fed into a linear classifier, $y = \text{softmax}(\mathbf{W}\mathbf{z} + \mathbf{b})$ to obtain the sentence label, where $\mathbf{W}$ and $\mathbf{b}$ are trainable parameters.

- **Sentence-Pair Classification.** For tasks like entailment detection, two sentence vectors, $\mathbf{z}_a$ and $\mathbf{z}_b$, are concatenated and passed to a linear classifier to obtain a label $y = \text{softmax}(\mathbf{W}[\mathbf{z}_a \| \mathbf{z}_b] + \mathbf{b})$, where $\|$ denotes channel-wise concatenation.

- **Similarity and Retrieval.** For ranking, the semantic relatedness of two vectors, $\mathbf{z}_a$ and $\mathbf{z}_b$, is measured with a function like cosine similarity, $\text{sim}(\mathbf{z}_a, \mathbf{z}_b) = \mathbf{z}_a^\top \mathbf{z}_b / \|\mathbf{z}_a\| \|\mathbf{z}_b\|$.

### 3.2 GLOT

We introduce GLOT, a trainable framework that transforms the token-level hidden states into a final, sentence-level vector, $\mathbf{z} = \text{GLOT}(\mathbf{X})$. As illustrated in Figure 2, this process involves three stages: (1) constructing a token graph, (2) refining token states with a graph neural network (GNN) denoted TOKEN-GNN, and (3) performing a learnable readout. Standard pooling methods treat the input sequence as a set of independent vectors. While computationally cheap, this independence assumption forces the model to discard inter-token dependencies during the compression step, making the representation susceptible to signal dilution from distractor tokens. GLOT challenges this paradigm by reframing pooling as *relational learning*. We hypothesize that the token space is not a set, but a latent graph $\mathcal{G}$, where nodes are tokens and edges represent semantic dependencies. This allows GLOT to learn complex, multi-token dependencies relevant to the task. This paradigm shift is significant because we are to the best of our knowledge the first to adapt the LLM's rich, yet possibly unoptimized, respect to sentence-level tasks, relational structure using a token-graph approach. Below we explain the core mechanism of GLOT:

**Step 1: Token Graph Construction.** Given token hidden states $\mathbf{X} = [\mathbf{x}_1, \mathbf{x}_2, \cdots, \mathbf{x}_L] \in \mathbb{R}^{L \times d}$ that are obtained from an LLM with hidden dimensionality $d$, after processing an input of length $L$, we construct a token graph $\mathcal{G} = (\mathcal{V}, \mathcal{E})$ where nodes $|\mathcal{V}| = L$ correspond to tokens. Edges are defined by the cosine similarity $\mathbf{S}_{ij}$ between token vectors $\boldsymbol{x}_i$ and $\boldsymbol{x}_j$. To induce a sparse, semantic structure, we only create edges where $\mathbf{S}_{ij}$ exceeds a threshold $\tau$, which is a hyperparameter, discussed in Section 4.

**Step 2: Refinement with TOKEN-GNN.** Next, we apply a lightweight graph neural network, dubbed TOKEN-GNN, to refine the token representations by modeling their interactions. With token hidden states $\mathbf{X}$, we initialize node features $\mathbf{H}^{(0)} = \mathbf{X}\,\mathbf{W}_{in} \in \mathbb{R}^{L \times p}$ using a learnable matrix $\mathbf{W}_{in} \in \mathbb{R}^{d \times p}$, where $p$ is the hidden dimension of the GNN. Overall, we apply $K$ GNN layers to produce a set of refined, structure-aware token representations $\mathbf{H}^{(K)} = \mathbf{U} = [\mathbf{u}_1, \cdots, \mathbf{u}_L] \in \mathbb{R}^{L \times p}$. Each layer $\ell = 1, \ldots, K$ of the TOKEN-GNN computes:

$$\mathbf{a}_i^{(\ell)} = \underset{j \in \mathcal{N}_i}{\text{AGGREGATE}} \left( \mathbf{h}_j^{(\ell)} \right) \in \mathbb{R}^p, \tag{1}$$

$$\mathbf{h}_i^{(\ell+1)} = \sigma \left( \mathbf{W}^{(\ell)} \text{CONCAT}(\mathbf{h}_i^{(\ell)}, \mathbf{a}_i^{(\ell)}) \right), \tag{2}$$

where $\mathbf{a}_i^{(\ell)}$ is the aggregated information from the neighbors $\mathcal{N}_i$ of token $i$, AGGREGATE is a permutation-equivariant aggregation function like sum or mean, $\mathbf{W}^{(\ell)} \in \mathbb{R}^{p \times 2p}$ is a learnable weight matrix, and $\sigma$ is a nonlinear activation function, with implementation details in Appendix B.

**Step 3: Readout Layer.** The set of refined token representations, $\mathbf{U}$, is aggregated into the sentence vector $\mathbf{z}$ via learnable scoring. A scalar importance score $m_i$ is computed for each refined token vector $\mathbf{u}_i$, normalized using softmax to create weights $\boldsymbol{\pi}$, and used to compute a weighted sum:

$$m_i = \mathbf{v}^\top \tanh(\mathbf{W}_m \mathbf{u}_i + \mathbf{b}_m), \quad \boldsymbol{\pi} = \text{softmax}(\mathbf{m}), \quad \mathbf{z} = \sum_{i=1}^{L} \pi_i \mathbf{u}_i, \tag{3}$$

where $\mathbf{m} = [m_1, \ldots, m_L]$.

Overall, GLOT aggregates token-level hidden states obtained from a frozen LLM, to obtain refined and learnable sentence-level representations by modeling token–token relationships using a graph and processing them using TOKEN-GNN.

**Properties of GLOT.** The GLOT framework extends several common methods for obtaining sentence-level representations, which can be recovered as special cases. If we disable the TOKEN-GNN by setting its number of layers to zero (i.e., $K = 0$), then the refined vectors are simply the original hidden states (that is, $\mathbf{u}_i = \mathbf{x}_i$), and the framework reduces to a direct weighted pooling mechanism. From here, we can model both standard pooling methods (like mean or CLS pooling) by using fixed weights and adaptive scoring methods, like AdaPool from Brothers (2025), by keeping the weights learnable.

These cases, where $K = 0$, fit into the DeepSets framework (Zaheer et al., 2017), in which all elements $\mathbf{x}_i$ are transformed individually $\phi(\mathbf{x}_i)$ before a global aggregation function. Instead, the Token-GNN utilized in GLOT enables information exchange in the form of $\phi(\mathbf{x}_i, \mathcal{G})$, taking a more global approach and allowing interactions between tokens. Bronstein et al. (2021) has shown DeepSets to be a special case of convolutional GNNs with no edge connectivity and, thus, strictly less powerful than message passing, an advantage we exploit in GLOT. The additional communication introduced in GLOT between tokens' representations allows it to model linguistic phenomena that hinge on pairwise or multi-hop dependencies among the tokens. The GNN mechanism in GLOT requires additional memory and computations, compared with pre-defined methods. Nonetheless, we note that, in comparison to other methods, which require the fine-tuning of the entire backbone LLMs, our GLOT strikes a balance between efficiency and effectiveness in downstream performance, as is evident in Section 4 and Figure 1.

## 4 EXPERIMENTS AND DISCUSSION

We conduct a comprehensive evaluation of GLOT to validate our core hypothesis: obtaining sentence-level representation via its reframing as relational learning before compression yields superior sentence embeddings from frozen LLMs compared with traditional and recent learnable approaches. Throughout our experiments, all backbone LLM models remain completely frozen; only the lightweight GLOT head and a minimal task-specific classifier are trained. This design ensures our approach is both parameter and resource-efficient. Our evaluation is guided by four key research questions:

Table 1: **A comparison of pooling methods on the GLUE benchmark using six different frozen backbones.** The table reports standard metrics: MCC for CoLA, Spearman for STS-B, F1 for MRPC/QQP, and Accuracy for the rest. Scores are multiplied by 100, with the **best** performance for each model highlighted in bold.

| Model | Method | CoLA MCC ↑ | SST-2 ACC ↑ | STS-B SPEA. ↑ | MRPC F1 ↑ | QQP F1 ↑ | MNLI-m ACC ↑ | MNLI-mm ACC ↑ | QNLI ACC ↑ | RTE ACC ↑ | WNLI ACC ↑ |
|---|---|---|---|---|---|---|---|---|---|---|---|
| BERT | [CLS] | 22.66 | 83.83 | 61.08 | 79.58 | 19.70 | 43.86 | 45.03 | 54.75 | 50.90 | 45.07 |
| | Mean | 19.55 | 82.91 | 74.96 | 80.28 | 29.01 | 43.86 | 45.16 | 56.43 | 51.62 | 52.11 |
| | Max | 15.79 | 80.73 | 74.12 | 81.64 | 29.58 | 38.60 | 39.55 | 53.79 | 51.98 | 49.26 |
| | AdaPool | 29.20 | 87.72 | 80.01 | 77.99 | 40.15 | 48.57 | 49.93 | 58.04 | 51.62 | 45.07 |
| | GLOT | **47.49** | **90.25** | **83.86** | **82.58** | **62.19** | **54.39** | **54.47** | **61.08** | **59.21** | **54.93** |
| RoBERTa | [CLS] | 6.92 | 66.63 | 52.87 | 81.22 | 47.66 | 32.78 | 32.98 | 54.89 | 52.34 | 40.85 |
| | Mean | 23.69 | 84.12 | 70.55 | 81.92 | 48.97 | 39.15 | 38.76 | 57.77 | 54.63 | 38.73 |
| | Max | 22.06 | 79.10 | 66.39 | 81.52 | 44.69 | 35.54 | 35.37 | 52.49 | 52.22 | 52.81 |
| | AdaPool | 26.80 | 90.97 | 71.12 | 80.78 | 57.71 | 42.51 | 44.24 | 59.72 | 50.45 | 41.90 |
| | GLOT | **56.08** | **92.78** | **85.27** | **81.95** | **61.41** | **57.01** | **57.95** | **62.73** | **56.68** | **56.34** |
| SmolLM2 | [EOS] | 7.63 | 77.75 | 52.77 | 81.03 | 38.11 | 41.14 | 42.66 | 53.23 | 49.10 | 47.88 |
| | Mean | 12.30 | 79.81 | 56.39 | 80.60 | 32.34 | 40.50 | 41.06 | 55.97 | 54.15 | 42.25 |
| | Max | 2.38 | 73.62 | 52.10 | 76.72 | 24.02 | 37.44 | 38.40 | 54.84 | 51.62 | 52.11 |
| | AdaPool | 7.21 | 83.71 | 61.20 | 81.69 | 49.26 | 41.00 | 42.35 | 58.08 | 55.59 | 45.07 |
| | GLOT | **39.23** | **90.25** | **76.28** | **82.24** | **62.32** | **53.42** | **53.64** | **59.86** | **57.40** | **63.38** |
| TinyLlama | [EOS] | 8.33 | 73.85 | 64.63 | 80.31 | 41.46 | 39.33 | 40.92 | 56.19 | 47.29 | 45.07 |
| | Mean | 5.93 | 73.85 | 61.29 | 80.67 | 41.46 | 39.50 | 40.83 | 57.51 | 49.58 | 45.07 |
| | Max | 2.76 | 70.87 | 63.99 | 81.45 | 39.64 | 36.88 | 37.93 | 55.29 | 50.90 | 46.48 |
| | AdaPool | 4.63 | 59.92 | 69.53 | 81.04 | 30.17 | 42.69 | 43.49 | 57.71 | 46.20 | 50.70 |
| | GLOT | **17.61** | **80.73** | **71.77** | **82.54** | **59.92** | **48.04** | **49.34** | **63.77** | **57.40** | **53.52** |
| LLaMA-3B | [EOS] | 37.37 | 91.74 | 74.11 | 70.58 | 58.78 | 48.47 | 47.46 | 53.98 | 54.87 | 42.25 |
| | Mean | 20.91 | 87.04 | 78.62 | 70.34 | 56.82 | 48.06 | 47.19 | 59.60 | 57.40 | 45.07 |
| | Max | 13.49 | 84.51 | 73.27 | 67.64 | 51.17 | 40.89 | 40.77 | 55.84 | 49.45 | 47.88 |
| | AdaPool | 43.32 | 92.54 | 81.93 | 71.81 | 49.37 | 49.56 | 50.59 | 58.48 | 55.23 | 47.88 |
| | GLOT | **55.13** | **93.92** | **82.83** | **82.34** | **61.16** | **53.49** | **54.67** | **67.15** | **61.01** | **56.34** |
| Mistral-7B | [EOS] | 38.63 | 92.55 | 72.36 | 76.32 | 51.68 | 48.18 | 48.33 | 50.82 | 50.90 | 40.85 |
| | Mean | 38.61 | 89.91 | 77.96 | 77.22 | 57.44 | 47.86 | 48.08 | 53.46 | 53.07 | 42.25 |
| | Max | 10.78 | 85.89 | 70.72 | 65.61 | 54.39 | 38.77 | 39.30 | 58.70 | 53.07 | 48.70 |
| | AdaPool | 48.00 | 93.00 | 79.55 | 81.12 | 49.07 | 50.72 | 51.56 | 55.75 | 54.87 | 49.30 |
| | GLOT | **54.30** | **94.38** | **80.51** | **82.83** | **64.07** | **51.66** | **53.22** | **60.93** | **59.21** | **56.34** |

(**RQ1**) How does GLOT compare to standard pre-defined and learnable sentence-level representation methods, across diverse LLMs and tasks?

(**RQ2**) Does explicit relational learning offer consistent improvements, especially for decoder-only models?

(**RQ3**) Can our GLOT match or exceed the performance of fine-tuned models while maintaining the computational efficiency of frozen LLMs?

(**RQ4**) How robust is GLOT to the signal dilution that affects traditional techniques?

## 4.1 EXPERIMENTAL SETUP

We evaluate GLOT against standard static (Mean, Max, CLS/EOS) and learnable pooling baselines across a diverse set of frozen encoder (BERT (Devlin et al., 2019), RoBERTa (Liu et al., 2019)) and decoder (e.g., Llama (Meta AI, 2024), Mistral (Jiang et al., 2023)) models. The evaluation is conducted on a wide range of tasks, including general language understanding (GLUE) (Wang et al., 2018), long-text classification (IMDB) (Maas et al., 2011), and retrieval (MTEB) (Muennighoff et al., 2023). To specifically test for relational robustness, we also introduce a synthetic diagnostic stress test that measures performance under noise. Across all experiments, the LLM backbones remain completely frozen. Full details on all models, baselines, benchmarks, training hyperparameters, and evaluation protocols are provided in Appendix B.

## 4.2 GENERAL LANGUAGE UNDERSTANDING EVALUATION (GLUE BENCHMARK)

Across the GLUE benchmark, GLOT consistently outperforms all baselines on all LLMs, from encoders like BERT to decoders like Mistral-7B. Table 1 provides the detailed scores, while Figure 4

Table 2: **Accuracy (×100) on the IMDB long-text sentiment classification task.** We freeze the LLM backbones and train only the pooling heads and a linear classifier. The **best** result per model is in bold.

| Method | BERT | RoBERTa | SmolLM2 | TinyLlama | LLaMA3.2-3B | Mistral-7B |
|---|---|---|---|---|---|---|
| [CLS]/[EOS] | 80.23 | 82.04 | 82.82 | 87.27 | 90.56 | 84.86 |
| Mean | 81.64 | 84.38 | 84.10 | 88.72 | 92.58 | 94.21 |
| Max | 60.78 | 58.80 | 63.41 | 75.45 | 80.90 | 64.43 |
| AdaPool | 85.45 | 90.91 | 91.56 | 92.61 | 95.71 | 95.66 |
| GLOT | **86.93** | **94.52** | **94.18** | **93.38** | **96.14** | **95.95** |

of Appendix C visualizes the overall trend, showing that our GLOT's advantage is consistent across different task categories. This directly addresses **(RQ1)** and **(RQ2)**.

GLOT achieves its most significant performance gains on tasks that require nuanced relational understanding. On the Corpus of Linguistic Acceptability (CoLA) (Warstadt et al., 2018), for instance, GLOT dramatically improves the Matthew's Correlation Coefficient for BERT by a relative improvement of **62.63**% and **13.13**% for Mistral-7B. This suggests that by explicitly modeling token relationships, our approach better captures the grammatical structure essential for this task. Similarly, on Quora Question Pairs **(QQP)**, a paraphrase detection task, GLOT delivers a large performance improvement margin over baselines for all tested architectures.

The consistent superiority on single-sentence classification (SST-2) (Socher et al., 2013b), semantic similarity (STS-B) (Agirre et al., 2007), and inference (RTE) (Dagan et al., 2006; Bar Haim et al., 2006; Giampiccolo et al., 2007; Bentivogli et al., 2009) tasks validates that our "relational learning before compression" approach yields more robust and general-purpose embeddings than methods that pool token states in isolation.

### 4.3   LONG-TEXT CLASSIFICATION

We assess performance on longer sequences using the IMDB dataset (Maas et al., 2011), where the task is to classify paragraph-length reviews. As shown in Table 2, GLOT consistently outperforms all baselines. For instance, it improves accuracy by nearly **4.5%** for RoBERTa over the strongest baseline and by an average of **+10.1%** relative improvement over the standard [EOS] token for decoder models. This result highlights the effectiveness of our graph-based approach on long-form text; unlike simple pooling, which can dilute sentiment signals across long contexts, GLOT's relational learning preserves and utilizes critical phrases for more accurate classification.

### 4.4   LARGE-SCALE BENCHMARKING ON MTEB

To assess GLOT's performance as a general-purpose sentence encoder, we evaluate it on seven diverse tasks from the Massive Text Embedding Benchmark (MTEB) (Muennighoff et al., 2023). Since many tasks are zero-shot, all learnable heads are first trained on the MS MARCO dataset (Bajaj et al., 2016) with a contrastive loss while keeping the LLM backbones frozen. The specific MTEB tasks are detailed in Appendix B.

The results in Table 3 show that GLOT is a robust performer across all tasks for both encoder- and decoder-only architectures. For RoBERTa, GLOT achieves the best score on all seven tested tasks, with a notable ×**3** improvement on SciFact. This advantage extends to decoders: with the Llama-3B backbone, GLOT secures a top performance of **0.5103 MAP** on AskUbuntuDupQuestions, rivaling strong encoder-only models. This strong general-purpose performance, achieved without expensive backbone fine-tuning, provides a clear affirmative answer to **(RQ3)**.

### 4.5   DIAGNOSTIC ANALYSIS: EVALUATING RELATIONAL ROBUSTNESS

To test for relational robustness under noise **(RQ4)**, we design a synthetic diagnostic task inspired by 'signal-in-noise' evaluations (Brothers, 2025) and the 'Needle in a Haystack' paradigm (Kamradt, 2023). The test involves injecting a short phrase containing a logical dependency (e.g.,

Table 3: **Zero-shot performance on seven diverse tasks from the MTEB benchmark.** Prior to evaluation, we train all learnable pooling heads on the MS MARCO dataset. The **best** performance for each frozen backbone is in bold.

| Model | Method | EmotionClass. ACC ↑ | SciFact NDCG@10 ↑ | RedditClust. V-Meas. ↑ | AskUbuntu MAP ↑ | STS12 Cos. Spea. ↑ | TwitterSemEval Max AP. ↑ | SummEval Cos. Spea. ↑ |
|---|---|---|---|---|---|---|---|---|
| BERT | [CLS] | 0.2412 | 0.0231 | 0.1417 | 0.4137 | 0.2153 | 0.3433 | 0.2792 |
|  | Mean | 0.3361 | 0.1769 | 0.2777 | 0.4584 | 0.3087 | 0.5613 | 0.2983 |
|  | Max | 0.2812 | 0.2771 | 0.2241 | 0.4553 | 0.3175 | 0.5450 | 0.3022 |
|  | AdaPool | 0.3513 | 0.2224 | 0.3403 | 0.4778 | 0.3941 | 0.5195 | 0.2918 |
|  | GLOT | **0.3715** | **0.2485** | **0.3630** | **0.5020** | **0.4862** | **0.5623** | **0.3068** |
| RoBERTa | [CLS] | 0.2759 | 0.0900 | 0.1908 | 0.4439 | 0.1667 | 0.4848 | 0.2347 |
|  | Mean | 0.2520 | 0.0825 | 0.1850 | 0.4621 | 0.3210 | 0.5456 | 0.2986 |
|  | Max | 0.2200 | 0.0116 | 0.1354 | 0.4491 | 0.2667 | 0.5000 | 0.2583 |
|  | AdaPool | 0.2135 | 0.0042 | 0.1475 | 0.4513 | 0.2026 | 0.4744 | 0.2276 |
|  | GLOT | **0.2909** | **0.2605** | **0.2184** | **0.4687** | **0.3688** | **0.5598** | **0.3083** |
| SmolLM2 | [EOS] | 0.2252 | 0.0012 | 0.1418 | 0.4113 | 0.1900 | 0.3613 | 0.2271 |
|  | Mean | 0.2396 | 0.1313 | 0.1708 | 0.4428 | 0.3824 | 0.4256 | 0.2335 |
|  | Max | 0.1923 | 0.0385 | 0.0960 | 0.4382 | 0.2458 | 0.3650 | 0.2530 |
|  | AdaPool | 0.2360 | **0.1702** | 0.1905 | 0.4461 | 0.4322 | 0.4153 | **0.2591** |
|  | GLOT | **0.2471** | 0.1834 | **0.2306** | **0.4529** | **0.4754** | **0.4343** | 0.2628 |
| TinyLlama | [EOS] | 0.2044 | 0.0042 | 0.0689 | 0.4275 | 0.1297 | 0.3532 | 0.2602 |
|  | Mean | 0.1898 | 0.0126 | 0.0687 | 0.4269 | 0.1633 | 0.3150 | 0.2450 |
|  | Max | 0.1820 | 0.0049 | 0.0591 | 0.4292 | 0.1842 | 0.3588 | 0.1178 |
|  | AdaPool | 0.2904 | 0.0602 | 0.1688 | 0.4004 | 0.0329 | 0.2811 | 0.2521 |
|  | GLOT | **0.2905** | **0.0916** | **0.1800** | **0.4341** | **0.2369** | **0.3804** | **0.2649** |
| LLaMA-3B | [EOS] | 0.2765 | 0.0087 | 0.1979 | 0.4420 | 0.2494 | 0.4141 | 0.1917 |
|  | Mean | 0.2920 | 0.4247 | 0.3034 | 0.4971 | 0.4296 | 0.4430 | 0.1924 |
|  | Max | 0.2478 | 0.4087 | 0.1943 | 0.4906 | 0.3367 | 0.4196 | 0.2347 |
|  | AdaPool | 0.2185 | 0.4140 | 0.2774 | 0.4946 | 0.3765 | 0.3216 | 0.2350 |
|  | GLOT | **0.3046** | **0.4586** | **0.3301** | **0.5103** | **0.4616** | **0.4431** | **0.2658** |
| Mistral-7B | [EOS] | 0.2662 | 0.0033 | 0.1858 | 0.4352 | 0.2307 | 0.3846 | 0.2042 |
|  | Mean | 0.2995 | 0.3735 | 0.2544 | 0.4774 | 0.3824 | 0.4106 | 0.1964 |
|  | Max | 0.2142 | 0.2116 | 0.1015 | 0.4577 | 0.3017 | 0.4151 | 0.2470 |
|  | AdaPool | 0.2832 | 0.4268 | 0.2398 | 0.4767 | 0.3641 | 0.3510 | 0.2346 |
|  | GLOT | **0.3016** | **0.4414** | **0.2623** | **0.4821** | **0.3905** | **0.4221** | **0.2774** |

`...not...keys...`) into a long sequence of random words. A binary classifier must then interpret the logic of the signal phrase, with difficulty controlled by increasing the distractor ratio from 20% to 90%. The pseudo-code for synthetic data generation is presented in Algorithm 2 of Appendix B.

The results in Figure 3 show a stark divergence. As noise increases, the accuracy of baseline methods collapses; on Mistral-7B, AdaPool's accuracy plummets from 92.2% to 78.4%, and Mean pooling drops to 63.8%. In contrast, GLOT remains robust, maintaining over **97%** accuracy even at the 90% distractor level. This confirms that GLOT's explicit token graph successfully bypasses the signal dilution that plagues methods reliant on global summary statistics. Full results are available in Table 7 in Appendix C.

## 4.6 Ablations and Analysis

We conduct a series of ablations and analyses to validate GLOT's design choices and quantify its computational efficiency.

**Impact of Graph Sparsity.** To understand the importance of constructing a well-formed semantic graph, we ablate the similarity threshold parameter, $\tau$, using the Mistral-7B backbone on GLUE benchmark. As shown in Table 4, the graph structure is critical to performance. When $\tau = 0.0$, the graph is fully connected, allowing noisy or irrelevant token relationships to dilute the message passing process, resulting in suboptimal performance on all tasks. As we increase $\tau$, pruning weaker edges, performance steadily improves across most tasks, plateauing in the range of $\tau = 0.4 - 0.6$. This confirms that not all token relations are equally important; by focusing on the strongest semantic connections via relational learning, GLOT produces a more robust sentence representation.

**Computational Efficiency.** To address **(RQ3)**, we compare the resource consumption of GLOT against full fine-tuning (Full FT) and Parameter-Efficient Fine-Tuning (PEFT) with LoRA (Hu et al., 2022). The results in Table 5 highlight the dramatic efficiency of our approach. Prior research indicates that catastrophic forgetting becomes increasingly severe when fine-tuning large-scale

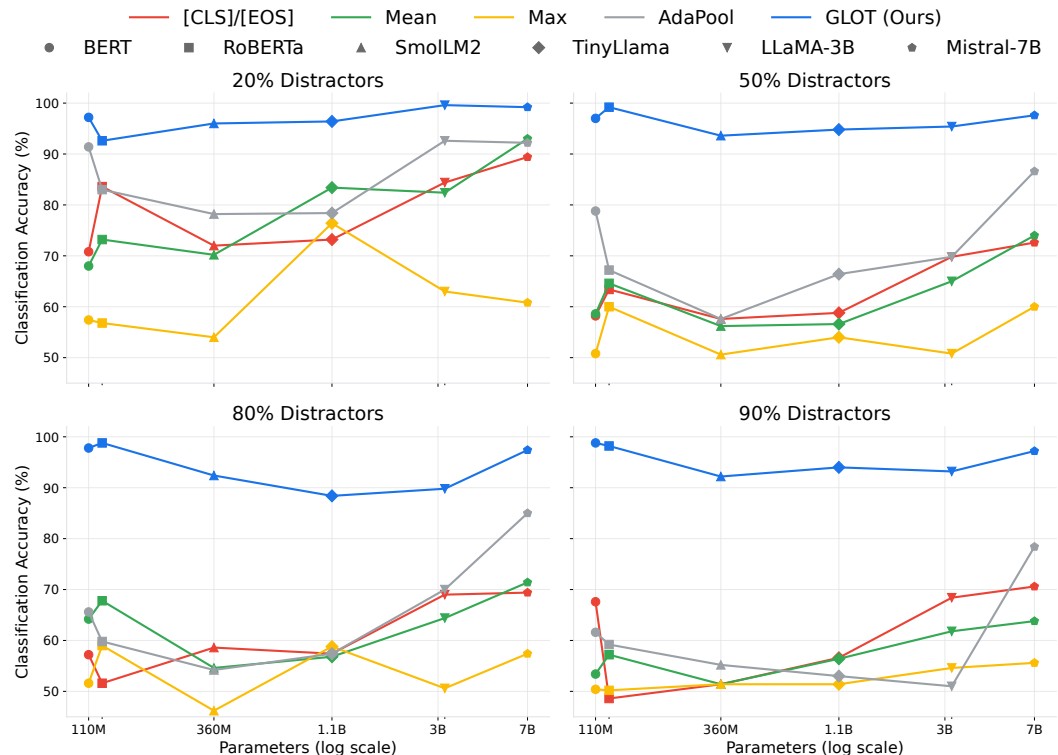

Figure 3: **Robustness to signal dilution on the diagnostic stress test.** Each of the four panels displays the classification accuracy for all pooling methods at a specific distractor ratio, which increases from 20% to 90%. Within each panel, backbone models are arranged along the x-axis by their parameter count.

Table 4: **An ablation study on the impact of graph sparsity in GLOT.** This table shows performance on GLUE tasks using the Mistral-7B backbone as we vary the similarity threshold ($\tau$) for token graph construction. All scores are multiplied by 100, and the **best** result for each task is in bold.

| Method | CoLA MCC ↑ | SST-2 ACC ↑ | STS-B SPEA. ↑ | MRPC F1 ↑ | QQP F1 ↑ | MNLI-m ACC ↑ | MNLI-mm ACC ↑ | QNLI ACC ↑ | RTE ACC ↑ | WNLI ACC ↑ |
|---|---|---|---|---|---|---|---|---|---|---|
| GLOT ($\tau = 0.0$) | 50.19 | 93.69 | 80.34 | 81.04 | 62.79 | 49.09 | 49.46 | 52.85 | 49.81 | 38.03 |
| GLOT ($\tau = 0.2$) | 53.40 | **94.38** | **80.48** | **82.83** | 62.53 | **51.66** | **53.22** | 54.15 | 49.45 | 36.62 |
| GLOT ($\tau = 0.4$) | 51.73 | 93.46 | 80.40 | 80.25 | **64.07** | 48.81 | 49.94 | **60.93** | 50.54 | 40.84 |
| GLOT ($\tau = 0.6$) | **54.30** | 93.23 | 80.29 | 80.06 | 63.49 | 49.36 | 50.01 | 53.67 | **54.15** | **56.34** |
| GLOT ($\tau = 0.8$) | 52.48 | 92.66 | 80.26 | 79.87 | 63.22 | 48.92 | 49.66 | 55.09 | 52.70 | **56.34** |

models on smaller downstream tasks (Li et al., 2024; Saroufim et al., 2025). The immense capacity of the 7B model makes it prone to overfitting the small training sets of GLUE tasks (like CoLA and RTE), degrading its generalizable representations. Furthermore, recent work suggests a functional equivalence between LoRA and Full Fine-Tuning (Shuttleworth et al., 2025), explaining why LoRA suffers from similar degradation. In contrast, GLOT requires only **8.92M** trainable parameters, which is approximately $20\times$ fewer than LoRA. This parameter efficiency translates directly to a minimal memory footprint of only **0.42 GB**, compared to over 32 GB for the other methods. Consequently, GLOT is over $100\times$ faster per training batch. This demonstrates that our method provides a practical and accessible way to generate high-quality embeddings from large, frozen LLMs on consumer-grade hardware. For a detailed breakdown of graph construction costs and extended cross-model benchmarks against BERT and Mistral, please refer to Appendix D.2.

Table 5: **A comparison of training methods by resource consumption and performance** on the CoLA task, using the Mistral-7B backbone. We contrast our frozen-backbone approach (GLOT) against full fine-tuning (Full FT) and LoRA. Batch runtime is reported as the mean $\pm$ standard deviation over 10 measurements.

| Method | # Trainable Params | GPU Memory (GB)↓ | Batch Runtime (ms)↓ | MCC↑ |
|---|---|---|---|---|
| Full FT + EOS | 7.11B | 32.59 | $1318.8 \pm 1.1$ | 49.63 |
| LoRA ($r = 64$) + EOS | 167.8M | 33.50 | $1454.6 \pm 1.1$ | 48.23 |
| GLOT (ours) | **8.92M** | **0.42** | $\mathbf{13.4 \pm 3.0}$ | **53.29** |

## 5 CONCLUSION

As LLMs continue to scale, the computational cost of full fine-tuning becomes prohibitive, establishing the need for improved pooling methods that operate on frozen backbones as a crucial research problem. In this work, we addressed a fundamental limitation of standard pooling: that it treats token hidden states as an independent set of vectors, discarding the rich relational structure captured by language models. We introduced GLOT, a lightweight and parameter-efficient pooling head that instantiates a new paradigm of relational learning followed by aggregation. GLOT first constructs a latent token-similarity graph, refines token representations using a GNN, and then aggregates them with an attention mechanism.

Through comprehensive experiments, we demonstrated that GLOT consistently outperforms strong baselines across a wide range of tasks and on both encoder- and decoder-only models. Our diagnostic stress test provided direct evidence that GLOT's graph-based learning makes it remarkably robust to the signal dilution that plagues traditional pooling. Furthermore, we showed that GLOT is up to two orders of magnitude more computationally efficient than parameter-efficient fine-tuning techniques like LoRA, making it a practical solution for adapting billion-parameter models.

Our findings challenge the view that pooling is a routine final step, showing instead that a carefully designed, relational learning-based head can unlock significant performance from frozen models. This work opens several avenues for future research, including exploring learnable graph construction mechanisms and applying the "relational learning before compression" paradigm to other modalities, such as pooling patch embeddings in Vision Transformers and designing advanced GNN architectures. Furthermore, our graph-based formulation opens the door to studying additional token interaction modeling techniques as future research work. For instance, recent work on graph rewiring (Barbero et al., 2024; Arnaiz-Rodríguez et al., 2022) and virtual nodes (Qian et al., 2024) have shown techniques in learning graph connectivity for graph learning tasks in GNNs. While GLOT is focused on introducing the concept of learning on token graphs already strong performance with cosine similarity, we view these dynamic rewiring strategies as an exciting avenue to further enhance the model's ability to capture long-range dependencies in future research works.

### ETHICS STATEMENT

Our work primarily focuses on developing a new pooling methodology and is evaluated on publicly available, standard academic benchmarks, including GLUE, MTEB, and IMDB. We do not use any private or sensitive user data, and our experiments did not involve human subjects. We acknowledge that the pre-trained language models used as backbones in our study may reflect societal biases present in their training corpora. Our proposed method, GLOT, operates on the outputs of these models and does not introduce new sources of bias, nor does it explicitly mitigate biases inherent in the backbone models. We intend for this work to contribute to the development of more efficient and robust NLP models, and we do not foresee any direct negative societal impacts from its application.

### REPRODUCIBILITY STATEMENT

To ensure the reproducibility of our results, we release our source code. Our methodology is described in Section 3, with detailed pseudo-code available in Algorithm 1. Appendix B provides a comprehensive description of our experimental setup, including the specific backbone models used, training and evaluation protocols, and all hyperparameters. All datasets used in our experiments are standard benchmarks publicly available through the Hugging Face Datasets library.

USAGE OF LARGE LANGUAGE MODELS (LLMS)

During the preparation of this manuscript, we utilized LLMs. Its role was strictly limited to that of grammatical assistance. The LLM was not used for research ideation, experimental design, data analysis, or the generation of any core scientific content. The authors take full responsibility for all content and claims presented in this paper.

ACKNOWLEDGMENTS

ZL acknowledges funding by the Federal Ministry of Research, Technology and Space of Germany and the state of North Rhine-Westphalia as part of the Lamarr Institute for Machine Learning and Artificial Intelligence. The authors gratefully acknowledge the access to the Marvin cluster of the University of Bonn, the computational cluster at Ben-Gurion University of the Negev, and Modal AI GPU credits. ME acknowledges support from the Israeli Ministry of Innovation, Science & Technology.

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

## A  ADDITIONAL RELATED WORK

**Fine-Tuning vs. Frozen Backbones for Embedding.**   A significant body of work adapts decoder-only LLMs into powerful text embedding models through extensive fine-tuning (Wang et al., 2023; Lee et al., 2025; Muennighoff et al., 2025; Ma et al., 2024; Tang & Yang, 2024). These methods achieve state-of-the-art performance but require modifying the LLM backbone, often through full-model training that is computationally prohibitive. GLOT sidesteps this entirely by operating on completely frozen backbones. Our approach is therefore lightweight, accessible, and applicable to both encoder-only and decoder-only models without expensive training.

**The Geometry of Embedding Space.**   Recent studies reveal that token embeddings from LLMs occupy anisotropic manifolds, which makes cosine similarity between pooled sentence vectors unreliable (Ethayarajh, 2019; Li et al., 2020). While post-processing methods like whitening can mitigate this (Su et al., 2021), they do not address the underlying information loss from pooling. SBERT-style fine-tuning reshapes this geometry but is computationally expensive. Our work offers an alternative: by constructing a similarity graph, GLOT operates on an approximation of the intrinsic manifold geometry, preserving relational structures that are lost when pooling in the ambient Euclidean space.

**Applications of Graph Neural Networks.**   The success of Graph Neural Networks (GNNs) is demonstrated by their wide-ranging application across numerous scientific and industrial domains. In the life sciences, they have become a cornerstone for molecular property prediction and drug discovery, where molecules are modeled as graphs of atoms and bonds (Gilmer et al., 2017; Xu et al., 2019). Similarly, they are used to analyze complex protein-protein interaction networks in bioinformatics. In the digital realm, GNNs power modern recommender systems by capturing the intricate relationships between users and items (Ying et al., 2018), and they are essential for learning over large-scale knowledge graphs (Schlichtkrull et al., 2018). Their foundational use case remains the analysis of social networks, where they are applied to tasks like node classification and community detection (Kipf & Welling, 2017; Hamilton et al., 2017). GNNs have also been successfully applied in other areas, including modeling particle systems in physics simulations (Sanchez-Gonzalez et al., 2020), processing 3D point clouds in computer vision, and solving complex combinatorial optimization problems like the Traveling Salesperson Problem (Cappart et al., 2023).

## B  IMPLEMENTATION DETAILS

### B.1  GENERAL SETUP

**Hardware and Software.**   All experiments were conducted on a single NVIDIA A6000 GPU. Our implementation is built using PyTorch (Paszke et al., 2019), with extensive use of the Hugging Face ecosystem (Wolf et al., 2020), including `transformers` for backbone models and `datasets` (Lhoest et al., 2021) for data loading. The graph-based components of our method are implemented using PyTorch Geometric (Fey & Lenssen, 2019). Large-scale benchmarking was performed using the `mteb` (Muennighoff et al., 2023) library, and retrieval metrics were calculated using `ranx`.

**Training Details.**   Unless otherwise noted, all trainable pooling heads were trained for 2 epochs using the Adam optimizer (Kingma & Ba, 2015) with a learning rate of $2 \times 10^{-4}$ and no weight decay. We used a training batch size of 32 and an evaluation batch size of 64. For all experiments, we used a fixed random seed of 42. To accelerate training, we implemented a feature to precompute and cache the frozen backbone's hidden states before training the pooling heads. We provide the pseudocode for GLOT in Algorithm 1. The hyperparameter tuning shown in Table 6 using Weights and Biases framework. To ensure a rigorous comparison in Table 5, we implemented Full Fine-Tuning (Full FT) and LoRA baselines using standard hyperparameters optimized for the Mistral-7B backbone. For Full Fine-Tuning, we fine-tuned the complete model on the training splits of CoLA, STS-B, and RTE for 3 epochs (Taori et al., 2023; Li et al., 2024). We used a learning rate of $2 \times 10^{-5}$ and a weight decay of $0.01$, utilizing the AdamW optimizer. For the LoRA baseline, we set the rank hyperparameter to $r = 64$ and applied adapters to both the attention and feed-forward blocks. The training used a higher learning rate of $2 \times 10^{-4}$ with a weight decay of $0.01$ for 3 epochs.

---

**Algorithm 1** GLOT: Graph-based Token Pooling

---

**Require:** $H \in \mathbb{R}^{B \times L \times d_{in}}$: Batch of hidden states from a frozen LLM.
$M \in \{0,1\}^{B \times L}$: Attention mask for the hidden states.
$\tau$: Cosine similarity threshold for edge creation.
$K$: Number of layers in the TOKEN-GNN.
**Ensure:** $Z \in \mathbb{R}^{B \times d_{out}}$: Batch of final sentence embeddings.

1: **function** GLOT($H, M$)
2:   $\mathcal{G}_{list} \leftarrow []$
3:   **for** $i = 1 \rightarrow B$ **do**          $\triangleright$ Step 1: Token Graph Construction
4:    $H'_i \leftarrow H[i, M[i] == 1, :]$       $\triangleright$ Get valid tokens for sentence $i$
5:    $S_i \leftarrow$ COSINESIMILARITY$(H'_i, H'_i)$     $\triangleright$ Pairwise similarity matrix
6:    $A_i \leftarrow (S_i > \tau)$       $\triangleright$ Create adjacency matrix based on threshold
7:    edge_index$_i \leftarrow$ ADJACENCYTOEDGES$(A_i)$
8:    $\mathcal{G}_{list}$.APPEND(nodes $= H'_i$, edges $=$ edge_index$_i$)
9:   **end for**
10:   $\mathcal{G}_{batch} \leftarrow$ BATCHGRAPHS$(\mathcal{G}_{list})$     $\triangleright$ Combine graphs into a single batch
11:   $U_0$, edge_index, batch_idx $\leftarrow \mathcal{G}_{batch}$.x, $\mathcal{G}_{batch}$.edge_index, $\mathcal{G}_{batch}$.batch
12:   $\mathcal{U}_{layers} \leftarrow [U_0]$
13:   **for** $k = 1 \rightarrow K$ **do**        $\triangleright$ Step 2: Refinement with TOKEN-GNN
14:    $U_{k-1} \leftarrow \mathcal{U}_{layers}[k-1]$
15:    $U_k \leftarrow$ GNN-LAYER$_k(U_{k-1}$, edge_index$)$
16:    $\mathcal{U}_{layers}$.APPEND$(U_k)$
17:   **end for**
18:   $U_{fused} \leftarrow$ JUMPINGKNOWLEDGECONCAT$(\mathcal{U}_{layers})$    $\triangleright$ Step 3: Feature Fusion
19:   $m \leftarrow$ READOUTMLP$(U_{fused})$      $\triangleright$ Step 4: Readout Layer
20:   $\pi \leftarrow$ SOFTMAXBYGRAPH$(m$, batch_idx$)$    $\triangleright$ Normalize scores per sentence graph
21:   $Z_{pooled} \leftarrow \pi \odot U_{fused}$        $\triangleright$ Apply attention weights
22:   $Z \leftarrow$ SCATTERADD$(Z_{pooled}$, batch_idx$)$   $\triangleright$ Aggregate via weighted sum per graph
23:   **return** $Z$
24: **end function**

---

Table 6: Hyperparameter search space for the GLOT pooling head. The final model configuration was determined via a grid search over these values. The search was performed consistently across all backbone models and datasets.

| Hyperparameter | Search Space |
|---|---|
| *Optimization* | |
| Learning Rate | {1e-3, 2e-4, 2e-5} |
| Weight Decay | {0.0, 1e-5, 5e-5} |
| | |
| *Token-GNN Architecture* | |
| GNN Layers ($K$) | {2, 4} |
| GNN Hidden Dimension | {64, 128, 256} |
| Jumping Knowledge | {cat, max, mean, none} |
| Input Projection Dimension | {128, 256, 512} |
| | |
| *Graph Construction* | |
| Similarity Threshold ($\tau$) | {0.1, 0.3, 0.6} |

## B.2 MODEL CONFIGURATIONS

**Backbone Models.** All backbone models were loaded from the Hugging Face Hub. For decoder-only models, the tokenizer's padding side was set to 'right'. If a model did not have a pre-defined padding token, the '[EOS]' token was used.

**Baseline Pooling Methods.** We implemented all baselines within the same framework and evaluated them to ensure a fair comparison.

- **Static Methods:** MEAN and MAX pooling operate over the non-padded token hidden states. [CLS]/[EOS] pooling for encoder models takes the hidden state of the first token. For decoder models, it takes the hidden state of the last non-padded token, identified via the attention mask.

- **AdaPool:** Our implementation follows the original paper (Brothers, 2025), consisting of a two-layer MLP with a Tanh activation that computes a scalar score for each token, followed by a softmax and weighted average.

**GLOT Configuration.** Our GLOT is implemented using 2 layers of GATConv (Veličković et al., 2018) with a hidden dimension of 128 and ReLU non-linearity (Nair & Hinton, 2010). As described in the main paper, the graph is constructed by creating edges between tokens where their cosine similarity exceeds a threshold of $\tau = 0.6$. Following the GNN layers, we use a 'cat' mode for Jumping Knowledge (Xu et al., 2018) to aggregate features from all layers before the final attention readout.

**Compatibility with Fine-Tuned Backbones** A natural question arises regarding whether GLOT provides complementary benefits when applied to a fine-tuned backbone rather than a frozen one. While GLOT is technically compatible with fine-tuned LLMs, our experimental results (refer to Table 17 and Table 1) demonstrate that applying GLOT to a *frozen* backbone already yields performance competitive and often superior to fully fine-tuned models (e.g., matching Full FT performance on CoLA).

Since our primary objective is to enable robust sentence representations without the prohibitive computational cost of updating billion-parameter backbones, we focused on the frozen setting. Furthermore, simultaneously fine-tuning the backbone while learning the graph structure introduces complex optimization dynamics that warrant a distinct study. We therefore consider the "Fine-tuned Backbone + GLOT" setting as a promising direction for future work.

## B.3 BENCHMARK-SPECIFIC DETAILS

**GLUE Benchmark.** For all tasks from the General Language Understanding Evaluation (GLUE) benchmark (Wang et al., 2018), we fine-tune the lightweight GLOT head and a task-specific linear classifier jointly on the training sets. Sequences are truncated to a maximum length of 128 tokens. For larger datasets (QQP, QNLI, MNLI), we train on a subsample of 20,000 examples.

**CoLA:** The Corpus of Linguistic Acceptability (Warstadt et al., 2018) requires the model to determine if a sentence is grammatically correct. **Task:** Binary classification. **Loss:** Cross-Entropy Loss.

**SST-2:** The Stanford Sentiment Treebank (Socher et al., 2013a) consists of movie reviews. **Task:** Binary sentiment classification (positive/negative). **Loss:** Cross-Entropy Loss.

**STS-B:** The Semantic Textual Similarity Benchmark (Agirre et al., 2007) involves predicting a similarity score between 1 and 5 for a pair of sentences. **Task:** Regression. **Loss:** Mean Squared Error (MSE) Loss.

**MRPC:** The Microsoft Research Paraphrase Corpus (Dolan & Brockett, 2005) contains sentence pairs. **Task:** Binary classification to determine if the sentences are paraphrases. **Loss:** Cross-Entropy Loss.

**QQP:** The Quora Question Pairs dataset requires determining if two questions are semantically equivalent. **Task:** Binary classification. **Loss:** Cross-Entropy Loss.

**MNLI:** The Multi-Genre Natural Language Inference corpus (Williams et al., 2018) provides a premise and a hypothesis. **Task:** Three-class classification (entailment, contradiction, neutral). **Loss:** Cross-Entropy Loss.

**QNLI:** The Question Natural Language Inference dataset, derived from SQuAD (Rajpurkar et al., 2016). **Task:** Binary classification to determine if a context sentence contains the answer to a question. **Loss:** Cross-Entropy Loss.

**RTE:** The Recognizing Textual Entailment datasets (Dagan et al., 2006; Bar Haim et al., 2006; Giampiccolo et al., 2007; Bentivogli et al., 2009). **Task:** Binary classification to determine if a premise entails a hypothesis. **Loss:** Cross-Entropy Loss.

**Long-Text Classification (IMDB).**   For the IMDB Large Movie Review dataset (Maas et al., 2011), sequences were truncated to a maximum length of 512 tokens. The dataset contains paragraph-length movie reviews. **Task:** Binary sentiment classification. **Loss:** Cross-Entropy Loss.

**MTEB Evaluation.**   For the Massive Text Embedding Benchmark (MTEB) (Muennighoff et al., 2023), we follow a two-stage process. First, the learnable pooling heads are trained on a large-scale retrieval dataset, and then they are evaluated in a zero-shot setting on the downstream MTEB tasks.

- **Training Stage:** All learnable heads were trained on the **MS MARCO** (Bajaj et al., 2016) passage ranking dataset. This involves predicting relevant text passages for a given query. **Task:** Passage retrieval. **Loss:** A symmetric in-batch contrastive loss with a temperature of 0.07.

- **Zero-shot Evaluation Stage:** The trained encoders are then evaluated on the following seven tasks without any further fine-tuning:

  - **EmotionClassification:** A multi-class classification task on tweets.
  - **SciFact:** A re-ranking task to verify scientific claims.
  - **RedditClustering:** An unsupervised task to cluster Reddit comments.
  - **AskUbuntuDupQuestions:** A retrieval task to find duplicate questions.
  - **STS12:** A semantic similarity regression task.
  - **TwitterSemEval2015:** A pair classification task for paraphrase detection.
  - **SummEval:** A summarization evaluation task based on semantic similarity.

### B.4   DIAGNOSTIC TASK GENERATION

The synthetic diagnostic task was created to isolate and test for relational understanding under noise.

- **Signal Phrases:** We created a small set of template phrases involving a logical dependency, such as negation (e.g., "The file has [X] but not [Y]").

- **Distractors:** The "haystack" was formed by sampling words randomly from a large general-purpose vocabulary derived from English Wikipedia.

- **Injection:** For each example, a 256-token sequence of random distractor words was generated. A signal phrase was then injected at a random position within this sequence.

- **Difficulty Control:** The difficulty was controlled by the **distractor ratio**, which we varied from 20% to 90%. A 90% distractor ratio means 90% of the tokens in the sequence are random noise.

The final dataset consists of 10,000 training examples and 2,000 test examples for each distractor ratio.

## C   ADDITIONAL RESULTS

**GLUE Benchmark.**   To provide a high-level summary of the comprehensive GLUE results presented in Table 1, we visualize the performance trends in Figure 4. To compare performance across different tasks and their associated metrics (e.g., Accuracy vs. MCC) on a single, unified scale, we normalized the scores. The methodology was as follows: for each of the eight GLUE tasks and for each of the six backbone models, we took the resulting scores of all five pooling methods and calculated their mean ($\mu$) and standard deviation ($\sigma$). Each individual score $x$ was then converted to a z-score via $z = (x - \mu)/\sigma$. These z-scores were then averaged within their respective task categories. A higher z-score indicates that a method's performance is significantly above the average of all tested methods for a given experimental setting. The plots clearly show that GLOT consistently achieves the highest z-score, often one or more standard deviations above the mean performance. This visualization powerfully reinforces our primary finding: the performance advantage of GLOT is not confined to specific tasks or model scales but is a robust and general phenomenon.

---

**Algorithm 2** Synthetic Diagnostic Dataset Generation

---

**Require:** $N$: Number of samples to generate.
**Require:** $L$: Total sequence length of each sample.
**Require:** $d_r$: The target distractor ratio (e.g., 0.2, 0.5, 0.8, 0.9).
**Require:** $\mathcal{T}$: A set of signal phrase templates, each with an associated label (e.g., '("...has [X] but not [Y]", 0)').
**Require:** $\mathcal{V}_D$: A large vocabulary of distractor words.
 1: **function** GENERATEDIAGNOSTICDATA($N, L, d_r, \mathcal{T}, \mathcal{V}_D$)
 2:      $\mathcal{D} \leftarrow \emptyset$                                    ▷ Initialize an empty dataset
 3:      $L_D \leftarrow \lfloor L \times d_r \rfloor$                          ▷ Calculate number of distractor tokens
 4:      $L_S \leftarrow L - L_D$                                   ▷ Calculate number of signal tokens
 5:      **for** $i = 1$ to $N$ **do**
 6:          $(template, label) \leftarrow$ RandomChoice($\mathcal{T}$)
 7:          $signal\_tokens \leftarrow$ Instantiate($template$)          ▷ e.g., fill placeholders like [X] and [Y]
 8:                                              ▷ Ensure signal phrase fits the allocated length
 9:          **if** length($signal\_tokens$) > $L_S$ **then**
10:              $signal\_tokens \leftarrow signal\_tokens[: L_S]$                     ▷ Truncate if too long
11:          **else**
12:              $padding \leftarrow L_S -$ length($signal\_tokens$)
13:              $signal\_tokens \leftarrow$ concat($signal\_tokens$, Sample($\mathcal{V}_D, padding$))          ▷ Pad with distractors if too short
14:          **end if**
15:          $distractor\_tokens \leftarrow$ Sample($\mathcal{V}_D, L_D$)          ▷ Sample distractors with replacement
16:          $p_{inject} \leftarrow$ RandomInt($0, L_D$)                    ▷ Choose a random injection point
17:          $sequence \leftarrow$ concat($distractor\_tokens[: p_{inject}], signal\_tokens, distractor\_tokens[p_{inject} :]$)
18:          $\mathcal{D} \leftarrow \mathcal{D} \cup \{(sequence, label)\}$
19:      **end for**
20:      **return** $\mathcal{D}$
21: **end function**

---

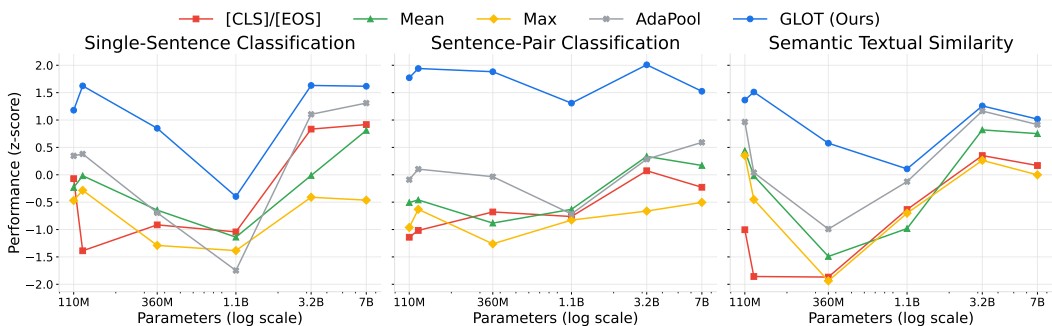

Figure 4: **Z-score normalized performance on the GLUE benchmark, aggregated by task category.** Performance, represented as a z-score, is plotted against the number of parameters in the frozen backbone model (log scale). A higher z-score indicates better relative performance compared to the average of all tested methods for that setting.

## C.1   DIAGNOSTIC TASK: DETAILED RESULTS AND ANALYSIS

To provide a controlled evaluation of relational robustness under noise (**RQ4**), we designed a synthetic diagnostic task. Inspired by 'signal-in-noise' evaluations (Brothers, 2025) and the 'Needle in a Haystack' paradigm (Kamradt, 2023), our stress test is specifically adapted to probe for relational understanding rather than simple factual recall. We programmatically generate sequences by injecting a short "signal phrase" with a logical dependency (e.g., negation) into a long sequence of random distractor words. The task is a binary classification based on the logic within the signal phrase. We

Table 7: **Full results for the diagnostic stress test**, which evaluates robustness to signal dilution. The table reports the classification accuracy for all pooling methods across six backbones as the ratio of distractor tokens in the input sequence increases from 20% to 90%. The **best** results for each model are in bold.

| Model | Method | 20% Distractors | 50% Distractors | 80% Distractors | 90% Distractors |
|---|---|---|---|---|---|
| BERT | [CLS] | 70.8 | 58.2 | 57.2 | 67.6 |
| | Mean | 68.0 | 58.6 | 64.2 | 53.4 |
| | Max | 57.4 | 50.8 | 51.6 | 50.4 |
| | AdaPool | 91.4 | 78.8 | 65.6 | 61.6 |
| | GLOT | **97.2** | **97.0** | **97.8** | **98.8** |
| RoBERTa | [CLS] | 83.6 | 63.4 | 51.6 | 48.6 |
| | Mean | 73.2 | 64.6 | 67.8 | 57.2 |
| | Max | 56.8 | 60.0 | 59.0 | 50.2 |
| | AdaPool | 83.0 | 67.2 | 59.8 | 59.2 |
| | GLOT | **92.6** | **99.2** | **98.8** | **98.2** |
| SmolLM2 | [CLS] | 72.0 | 57.6 | 58.6 | 51.4 |
| | Mean | 70.2 | 56.2 | 54.6 | 51.4 |
| | Max | 54.0 | 50.6 | 46.2 | 51.4 |
| | AdaPool | 78.2 | 57.6 | 54.2 | 55.2 |
| | GLOT | **96.0** | **93.6** | **92.4** | **92.2** |
| TinyLlama | [EOS] | 73.2 | 58.8 | 57.4 | 56.6 |
| | Mean | 83.4 | 56.6 | 56.8 | 56.4 |
| | Max | 76.4 | 54.0 | 58.8 | 51.4 |
| | AdaPool | 78.4 | 66.4 | 57.4 | 53.0 |
| | GLOT | **96.4** | **94.8** | **88.4** | **94.0** |
| LLaMA-3B | [EOS] | 84.4 | 69.8 | 69.0 | 68.4 |
| | Mean | 82.4 | 65.0 | 64.4 | 61.8 |
| | Max | 63.0 | 50.8 | 50.6 | 54.6 |
| | AdaPool | 92.6 | 69.8 | 70.0 | 51.0 |
| | GLOT | **99.6** | **95.4** | **89.8** | **93.2** |
| Mistral-7B | [EOS] | 89.4 | 72.6 | 69.4 | 70.6 |
| | Mean | 93.0 | 74.0 | 71.4 | 63.8 |
| | Max | 60.8 | 60.0 | 57.4 | 55.6 |
| | AdaPool | 92.2 | 86.6 | 85.0 | 78.4 |
| | GLOT | **99.2** | **97.6** | **97.4** | **97.2** |

systematically increase the task's difficulty by increasing the distractor ratio from 20% to 90%. The full generation process is detailed in Algorithm 2.

The complete results for this stress test are presented in Table 7. The data provides a clear and quantitative confirmation of our hypothesis: GLOT's performance remains remarkably stable even at extreme noise levels, while the performance of all baseline methods degrades significantly as the signal is diluted.

This trend is consistent across all architectures. For the encoder-only BERT backbone, GLOT's accuracy remains consistently above 97% across all distractor ratios. In contrast, the next-best baseline, AdaPool, sees its performance drop sharply from 91.4% at 20% distractors to just 61.6% at 90% distractors. The pattern is mirrored in decoder-only models. With the Llama backbone, GLOT's accuracy is nearly perfect at low noise (99.6%) and stays high at 83.2% even at the extreme 90% distractor ratio. All other methods, including using the standard '[EOS]' token, see their performance collapse, with most falling to near-chance levels. This analysis demonstrates that by explicitly modeling token relationships, GLOT can reliably identify and reason over the crucial signal phrase, whereas methods that rely on global summary statistics are overwhelmed by the distractor tokens.

Table 8: **Impact of Graph Topology ($\tau$) on Performance and Linearity.** We compare the sensitivity of BERT (top) and Mistral (bottom) to the threshold $\tau$. *Linear Probing* indicates the accuracy of a linear SVM trained on frozen GLOT embeddings, serving as a proxy for linear separability.

| Threshold ($\tau$) | STSB (Spear. ↑) | CoLA (MCC ↑) | RTE (ACC ↑) | Linear Probing (ACC ↑) |
|---|---|---|---|---|
| *BERT (Encoder-only)* | | | | |
| $\tau = 0.0$ | 81.88 | 39.62 | 50.90 | 76.41 |
| $\tau = 0.2$ | 82.12 | 39.82 | 50.90 | 76.51 |
| $\tau = 0.4$ | 82.25 | **47.49** | 52.34 | 77.18 |
| $\tau = 0.6$ | **83.86** | 43.16 | **59.21** | **77.56** |
| $\tau = 0.8$ | 83.85 | 43.16 | 52.70 | 77.18 |
| *Mistral-7B (Decoder-only)* | | | | |
| $\tau = 0.0$ | 80.34 | 49.81 | 50.90 | 80.06 |
| $\tau = 0.2$ | **80.48** | 49.45 | 50.90 | 81.20 |
| $\tau = 0.4$ | 80.40 | 50.54 | 52.34 | 80.44 |
| $\tau = 0.6$ | 80.29 | **54.15** | **59.21** | **81.40** |
| $\tau = 0.8$ | 80.26 | 52.70 | 52.70 | 80.82 |

## D  ADDITIONAL ANALYSES

### D.1  GRAPH TOPOLOGY AND REPRESENTATION QUALITY ANALYSIS

In this subsection, we expand upon the graph construction analysis presented in Table 4. To address the question of how graph topology influences representation quality beyond simple ablation, we conduct a detailed sensitivity analysis of the sparsity threshold $\tau$ on both encoder-only (BERT) and decoder-only (Mistral) backbones.

We evaluate performance across three semantically distinct tasks from GLUE (STSB, CoLA, and RTE). Furthermore, to provide a theoretical justification for the representation quality, we perform a **linear probing experiment**. In this setting, we freeze the sentence embeddings $z$ generated by GLOT (with no classifier head trained) and train a standard linear SVM. The resulting accuracy serves as a direct quantitative measure of the linear separability of the pooled embeddings.

As shown in Table 8, we observe two key trends:

1. **Correlation with Linear Separability:** The threshold $\tau$ that yields the highest linear probing accuracy (typically $\tau = 0.6$) consistently aligns with the highest performance on downstream tasks (RTE and STSB). This suggests that the graph structure explicitly refines the manifold of the embeddings, making classes more linearly separable.

2. **Task-Dependent Topology:** Different tasks benefit from different levels of sparsity. For example, CoLA (linguistic acceptability) on BERT peaks at $\tau = 0.4$, while RTE (entailment) benefits from a sparser graph at $\tau = 0.6$. This confirms that the thresholding mechanism allows GLOT to adapt the graph topology to the specific semantic or syntactic needs of the task.

### D.2  DETAILED EFFICIENCY AND SCALABILITY ANALYSIS

To fully analyze the computational cost (RQ3) and scalability to long contexts (RQ4), we provide extended benchmarks covering cross-model performance and a stress test of the graph construction step.

**Cross-Model and Cross-Task Efficiency**  In Table 9, we compare GLOT against Full Fine-Tuning (Full FT) and LoRA ($r = 64$) across both encoder-only (BERT) and decoder-only (Mistral) architectures. We report performance on three semantically diverse tasks: CoLA (Grammar), STS-B (Similarity), and RTE (Entailment).

GLOT achieves a superior trade-off across both architectures:

Table 9: **Cross-Model Efficiency Benchmarks.** We compare GLOT against Full Fine-Tuning and LoRA. **MCC**: CoLA, **Spear.**: STSB, **ACC**: RTE. Runtimes are measured with batch size 32. GLOT provides consistent efficiency gains across architectures.

| Method | Trainable Params | GPU Mem. (GB) ↓ | Runtime (ms) ↓ | CoLA (MCC) ↑ | STSB (Spear.) ↑ | RTE (ACC) ↑ |
|---|---|---|---|---|---|---|
| | | *Mistral-7B (Decoder-only)* | | | | |
| Full FT + [EOS] | 7.11B | 32.59 | $1318.8 \pm 1.1$ | 49.63 | 55.68 | 55.23 |
| LoRA ($r = 64$) | 167.8M | 33.50 | $1454.6 \pm 1.1$ | 48.23 | 54.54 | 53.43 |
| GLOT (Ours) | **8.92M** | **0.42** | $\mathbf{13.4 \pm 3.0}$ | **53.29** | **80.51** | **59.21** |
| | | *BERT (Encoder-only)* | | | | |
| Full FT + [CLS] | 109.5M | 0.74 | $52.5 \pm 0.1$ | 38.31 | 60.10 | 56.68 |
| LoRA ($r = 64$) | 10.7M | 0.86 | $77.4 \pm 0.2$ | 36.16 | 64.64 | 56.32 |
| GLOT (Ours) | **8.92M** | **0.42** | $\mathbf{13.4 \pm 3.0}$ | **47.49** | **83.86** | **59.21** |

Table 10: **Scalability Stress Test.** Graph construction time vs. total inference time (per sample) at maximum context lengths ($L_{max}$). All times are in milliseconds (ms). Even at $L = 32K$, the graph construction overhead is $\approx 1.3\%$ of the total runtime.

| Backbone | Max Context ($L_{max}$) | Graph Const. (ms) | Total Runtime (ms) | Overhead (%) |
|---|---|---|---|---|
| BERT | 512 | $0.043 \pm 0.002$ | $5.36 \pm 0.06$ | 0.8% |
| TinyLlama-1.1B | 2048 | $0.672 \pm 0.001$ | $143.15 \pm 0.05$ | 0.5% |
| SmolLM2 | 8192 | $4.46 \pm 0.05$ | $772.30 \pm 0.11$ | 0.6% |
| Llama-3B | 8192 | $15.05 \pm 0.37$ | $2041.77 \pm 0.19$ | 0.7% |
| Mistral-7B | 32768 | $303.47 \pm 1.02$ | $23460.29 \pm 4.77$ | 1.3% |

- **Efficiency:** On Mistral-7B, GLOT reduces memory usage from $\approx 32$GB (Full FT) to just 0.42GB and reduces batch runtime from $\approx 1318$ms to 13.4ms ($100\times$ speedup).

- **Consistency:** The performance gains are not isolated to specific tasks; GLOT outperforms parameter-heavy baselines on all three benchmarks, confirming that the graph-based approach generalizes well across different semantic objectives.

**Scalability to Long Contexts** A theoretical concern with graph-based pooling is the $\mathcal{O}(L^2)$ complexity of edge formation, which could potentially become a bottleneck for long sequences. To investigate this, we benchmarked the graph construction time against the total forward pass runtime across the maximum supported context lengths of our backbone models (up to 32K tokens for Mistral-7B).

As shown in Table 10, even at extreme lengths, the graph construction overhead remains a small fraction of the total inference time. For instance, with Mistral-7B at a context length of 32,768 tokens, graph construction takes $\approx 0.3$ seconds, which is negligible compared to the computational cost of the backbone's forward pass ($\approx 23.5$ seconds). This confirms that the $\mathcal{O}(L^2)$ step does not hinder scalability in practical long-context applications.

### D.3 EFFECT OF GNN BACKBONE ARCHITECTURE

To evaluate the robustness of the GLOT framework and verify that our performance gains stem from the graph-based paradigm rather than a specific architecture, we experimented with different GNN backbones. In addition to GAT (Veličković et al., 2018) used in the main experiments, we evaluated using GCN (Kipf & Welling, 2017) abd GIN (Xu et al., 2019) architectures.

Table 11 presents the comparative results for Mistral-7B and BERT, respectively. The results yield two key observations: (i) **Robustness of the Paradigm:** Consistently across both LLM backbones (Mistral and BERT), all graph-based variants (GCN, GAT, GIN) significantly outperform the set-based AdaPool baseline. This validates our core hypothesis that modeling inter-sample relationships is critical for performance. (ii) **Architecture Sensitivity:** The optimal GNN architecture appears to

Table 11: **Performance comparison of different GNN backbones.** We evaluate the effectiveness of different graph variants against the AdaPool baseline using both Mistral-7B and BERT embeddings.

| Method | CoLA (MCC) | STSB (Spearman) | RTE (ACC) |
|---|---|---|---|
| *Mistral-7B (Decoder-only)* | | | |
| AdaPool (No GNN) | 48.00 | 79.55 | 54.87 |
| GLOT (GCN) | 52.65 | 79.74 | 57.04 |
| GLOT (GAT) (Ours) | 54.30 | **80.51** | 59.21 |
| GLOT (GIN) | **59.30** | 79.73 | **59.30** |
| *BERT (Encoder-only)* | | | |
| AdaPool (No GNN) | 29.20 | 80.01 | 51.62 |
| GLOT (GCN) | 45.19 | 80.17 | 58.12 |
| GLOT (GAT) (Ours) | 47.49 | **83.86** | **59.21** |
| GLOT (GIN) | **47.78** | 80.71 | 57.04 |

depend on the underlying LLM embeddings. For the larger Mistral-7B model, the more expressive GIN outperforms if not competitive. However, for BERT, GAT remains the superior choice.

### D.4 Comparison with Prompting-based Methods

Our goal in this work is to treat token embeddings for a given sentence from frozen LLMs as a semantic graph in the latent space. We reframe pooling as relational learning from token interactions, rather than treating tokens as a set of independent vectors. Orthogonally, prompting methods use hand-crafted prefixes (which are identical across all sentences and datasets) to obtain a representation for the sentence.

**Our goal is not to introduce a new memory-efficient pooling mechanism.** The computational efficiency of GLOT is a consequence of keeping the backbone frozen, thereby eliminating the requirement to store backbone gradients in memory and allowing the forward pass (to obtain token embeddings) to be amortized as a dataset preparation step.

Nevertheless, to contextualize our performance, we compare GLOT to prompting-based approaches, including PromptBERT (Jiang et al., 2022), PromptEOL (Jiang et al., 2024), and Pretended Chain of Thought and Knowledge Enhancement (Zhang et al., 2024). Table 13 presents the results on the STS-B benchmark.

As shown, GLOT consistently outperforms prompting methods on both encoder and decoder architectures. These results confirm that explicitly learning relational structures over tokens is more effective than input prompting for frozen LLMs.

### D.5 Comparison with Contrastive Fine-tuning Baselines

We acknowledge the importance of established sentence embedding baselines such as Sentence-BERT (Reimers & Gurevych, 2019), SimCSE (Gao et al., 2021), and recent adaptation methods like LLM2Vec (BehnamGhader et al., 2024). A key distinction, however, is that these methods require updating the backbone model (or altering attention mechanisms), whereas GLOT operates strictly on a **frozen backbone**.

To illustrate the performance-efficiency trade-off, Table 14 compares GLOT (using a frozen BERT backbone) against fully fine-tuned SBERT (MPNet-v2) and SimCSE (BERT-sup) on three linguistically diverse GLUE tasks. As shown, GLOT outperforms the fully fine-tuned baselines on tasks requiring complex linguistic understanding, such as CoLA (+22.5 points) and RTE (+6.5 points). While SimCSE retains a slight edge on semantic similarity (STS-B), GLOT remains highly competitive (within 3.4 points) despite having approximately $12\times$ fewer trainable parameters and requiring significantly less training time.

### D.6    IMPACT OF RELATIONAL LEARNING VS. MODEL CAPACITY

To verify that the performance gains of GLOT stem from its graph-based design rather than simply having more learnable parameters than standard pooling methods, we compared GLOT against a parameter-matched baseline. Specifically, we replaced the TOKEN-GNN module with a deep Multi-Layer Perceptron (MLP) of comparable capacity (9.2M parameters) and evaluated it using the Mistral-7B backbone.

Table 15 presents the results on three diverse GLUE tasks. As shown, GLOT consistently outperforms the parameter-matched MLP baseline across all tasks, despite using slightly fewer parameters (8.9M vs. 9.2M). The performance gap is most pronounced on STS-B (+6.4 points), where modeling fine-grained semantic similarity is critical. This difference highlights a fundamental limitation of the MLP, which processes the input tokens as a static vector. In contrast, GLOT employs a GNN over the token graph, enabling tokens (nodes) to explicitly exchange information via message passing. This confirms that the superior performance of GLOT is driven by *graph construction and relational learning*, not merely by learnable parameter capacity.

### D.7    INFERENCE-TIME COMPUTATIONAL COSTS

We benchmark the inference-time costs of GLOT (including graph construction and the GNN forward pass) against simpler pooling methods using the Mistral-7B backbone. We also report performance on representative datasets from the GLUE benchmark (CoLA for MCC, STS-B for Spearman, and RTE for Accuracy).

Table 16 summarizes these results. As observed, the inference time is dominated by the forward pass through the large LLM backbone. Consequently, the inference-time costs are nearly identical across all pooling methods. GLOT requires only $\approx 600$ MB of additional GPU memory and negligible additional runtime ($\approx 3$ms) compared to the baselines. This efficiency is achieved through specialized sparse computation operations utilized in our implementation.

For a detailed breakdown of the specific graph construction overhead across varying backbones and context lengths, please refer to Table 10. These results collectively indicate that GLOT offers a highly efficient pooling mechanism for frozen LLMs, providing significant performance improvements with minimal computational overhead.

### D.8    UNIFIED COMPARISON OF EFFICIENCY AND PERFORMANCE

To provide a holistic view of the trade-offs between computational resources and downstream effectiveness, we present a unified comparison using the Mistral-7B backbone. Table 17 contrasts training efficiency (parameters and memory) against performance on three diverse tasks: CoLA (linguistic acceptability), STS-B (semantic similarity), and RTE (textual entailment).

As shown, GLOT introduces only a minor parameter increase compared to AdaPool (8.9M vs. 2.1M) yet yields significant gains across all metrics (e.g., +5.3 points on CoLA and +4.3 points on RTE). Furthermore, GLOT consistently outperforms the parameter-matched MLP baseline, confirming the value of the relational graph structure.

Most notably, when compared to fine-tuning approaches, GLOT outperforms both LoRA and Full Fine-Tuning on all three tasks. It achieves this while requiring $\approx 19\times$ fewer parameters than LoRA and $\approx 800\times$ fewer than Full FT, utilizing only a fraction of the GPU memory. This confirms that GLOT offers a "sweet spot," delivering performance competitive with (or superior to) fine-tuning techniques while maintaining the computational efficiency of frozen methods.

### D.9    VISUALIZING TOKEN CONTRIBUTIONS

To understand why graph-based pooling yields superior representations compared to set-based pooling or average pooling, we visualize the token contribution weights ($\pi$) assigned by different methods. Figure 5 illustrates the weight distribution for Mean Pooling, AdaPool (Brothers, 2025) and GLOT on samples from Quora Question Pairs dataset using a frozen BERT backbone. The visualization highlights a distinct difference in how these pooling methods prioritize information:

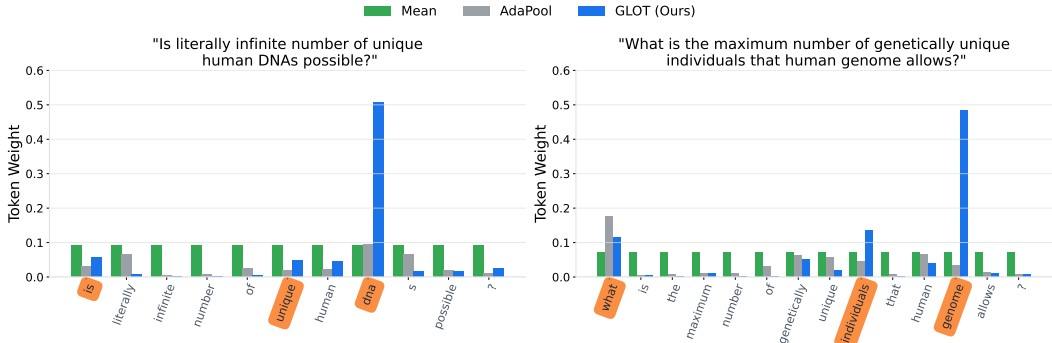

(a) Example 1: GLOT (blue) focuses on "DNA", "genome", and "individuals", while suppressing the interrogative "What".

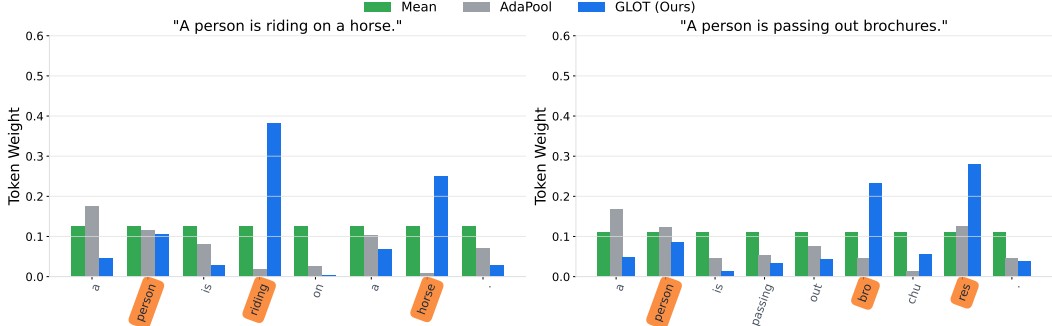

(b) Example 2: GLOT isolates the action "riding" and object "horse", whereas AdaPool (grey) often attends to functional stop words like "a" or "person".

Figure 5: **Token Contribution Analysis with frozen BERT.** Visualization of learned token weights ($\pi$) on 2 examples. The **orange highlights** on the X-axis indicate the top-3 scoring tokens identified by GLOT. While Mean Pooling (green) is uniform and AdaPool (grey) tends to over-index on high-frequency functional words, GLOT (blue) consistently up-weights the semantic anchors essential for determining sentence equivalence.

- **Mean Pooling** assigns a uniform weight of $(1/L)$ to all tokens. This approach suffers from a signal dilution, as tokens without meaning (e.g. 'is','of', '?') contribute equally to the final representation as the important tokens.

- **AdaPool** weighs tokens non-uniformly and treats tokens independently. We observe that it frequently assigns high importance to functional words, syntactic markers or interrogatives (e.g., attending to 'What' or 'a'). This suggests the method is overfitting to common patterns rather than semantic token interactions.

- Our **GLOT** exhibits a highly selective distribution. By refining the representations via the TOKEN-GNN before aggregation, GLOT identifies and assigns larger weights to semantically important tokens of the sentence. For example, in the query "What is the maximum number of genetically unique individuals that human genome allows?", GLOT suppresses the interrogative "What" and places maximum weight on "genome" and "individuals". Similarly, in "A person is riding a horse", GLOT isolates the action "riding" and the object "horse", whereas AdaPool focuses on the article "a".

This analysis suggests that the graph structure enables GLOT to perform relational learning by exchanging information among the tokens thereby producing a robust embedding that is resilient to the distractors inherent in natural language.

Table 12: Comprehensive MTEB benchmark results. Datasets are grouped by task category, with the category average reported in the shaded rows. The evaluation metric for each category is indicated in parentheses: CLASSIFICATION (Accuracy), RETRIEVAL (NDCG@10), CLUSTERING (V-Measure), STS (Spearman Correlation), RERANKING (MAP), PAIRCLASSIFICATION (Average Precision), and SUMMARIZATION (Spearman Correlation). †Evaluated in half precision due for datasets with > 100K sentences, due to resource constraints. **First**, **Second**, **Third**.

| Dataset | [EOS] | Mean | MaxPool | AdaPool | GLOT |
|---|---|---|---|---|---|
| CLASSIFICATION (ACC ↑) | 0.4910 | 0.5178 | 0.3917 | 0.4907 | 0.4727 |
| AmazonCounterfactualClassification | 0.5767 | **0.6301** | 0.6052 | **0.6076** | **0.6210** |
| AmazonPolarityClassification | **0.7858** | **0.6912** | **0.6447** | 0.6364 | 0.6299 |
| AmazonReviewsClassification | **0.3865** | **0.3544** | 0.2842 | **0.3068** | 0.3048 |
| Banking77Classification | 0.3127 | **0.4482** | 0.2704 | **0.4425** | **0.4730** |
| EmotionClassification | 0.2662 | **0.2995** | 0.2142 | **0.2832** | **0.3016** |
| ImdbClassification | 0.6209 | **0.7214** | 0.5982 | **0.6807** | **0.6704** |
| MTOPDomainClassification | 0.5783 | **0.6596** | 0.4323 | **0.6102** | **0.6103** |
| MTOPIntentClassification | 0.3036 | **0.4757** | 0.2465 | **0.3980** | **0.3970** |
| MassiveIntentClassification | **0.4265** | 0.3633 | 0.2083 | **0.4090** | **0.4254** |
| MassiveScenarioClassification | **0.5114** | 0.4374 | 0.2483 | **0.4707** | **0.4963** |
| ToxicConversationsClassification | **0.6446** | **0.6412** | 0.5347 | 0.5922 | 0.5950 |
| TweetSentimentExtractionClassification | **0.4808** | **0.4923** | 0.4150 | **0.4518** | 0.4495 |
| RETRIEVAL (NDCG@10 ↑) | 0.0558 | 0.1227 | 0.0809 | 0.1780 | 0.1759 |
| ArguAna | 0.0875 | **0.4162** | 0.0986 | **0.3164** | **0.3003** |
| CQADupstackRetrieval | 0.0106 | **0.0926** | 0.0291 | **0.0954** | **0.1143** |
| ClimateFEVER† | 0.0188 | **0.0456** | 0.0177 | **0.0842** | **0.0955** |
| DBPedia† | **0.0121** | **0.0121** | 0.0095 | **0.0788** | **0.0857** |
| FEVER† | 0.0168 | **0.0532** | 0.0303 | **0.1357** | **0.1451** |
| FiQA2018 | 0.0111 | **0.0333** | 0.0123 | **0.0840** | **0.0953** |
| HotpotQA† | 0.0100 | 0.0361 | **0.0387** | **0.2251** | **0.2096** |
| NFCorpus | 0.0233 | **0.0253** | 0.0146 | **0.0925** | **0.1043** |
| NQ† | 0.0064 | **0.0065** | 0.0036 | **0.0870** | **0.0860** |
| QuoraRetrieval | **0.5697** | 0.5332 | 0.5200 | **0.6030** | **0.6055** |
| SCIDOCS | 0.0035 | 0.0181 | **0.0187** | **0.0441** | **0.0480** |
| SciFact | 0.0033 | **0.3735** | 0.2116 | **0.4268** | **0.4414** |
| TRECCOVID | 0.0581 | 0.1706 | **0.1840** | **0.2643** | **0.2626** |
| Touche2020 | 0.0037 | **0.0207** | **0.0217** | **0.0935** | 0.0037 |
| MSMARCO† | 0.0022 | 0.0034 | **0.0040** | **0.0389** | **0.0415** |
| CLUSTERING (V-MEAS. ↑) | 0.2254 | 0.3245 | 0.2197 | 0.3312 | 0.2995 |
| ArxivClusteringP2P† | 0.2979 | 0.2979 | **0.3791** | **0.4691** | **0.4548** |
| ArxivClusteringS2S | **0.2770** | **0.3009** | 0.1722 | **0.2858** | 0.2466 |
| BiorxivClusteringP2P | 0.1415 | **0.3588** | 0.2503 | **0.3593** | **0.3484** |
| BiorxivClusteringS2S | 0.1341 | **0.2336** | 0.1082 | **0.2049** | **0.1926** |
| MedrxivClusteringP2P | 0.1478 | **0.3018** | 0.2516 | **0.3140** | **0.3110** |
| MedrxivClusteringS2S | 0.1664 | **0.2414** | 0.1616 | **0.2287** | **0.2201** |
| RedditClustering | 0.1858 | **0.2544** | 0.1015 | **0.2398** | **0.2623** |
| RedditClusteringP2P | 0.2997 | **0.5755** | 0.4121 | **0.5553** | 0.2997 |
| StackExchangeClustering | **0.4214** | **0.4523** | 0.2215 | **0.4183** | 0.4116 |
| StackExchangeClusteringP2P | 0.2280 | **0.3522** | 0.2594 | **0.3493** | **0.3333** |
| TwentyNewsgroupsClustering | 0.1804 | **0.2007** | 0.0992 | **0.2189** | **0.2149** |
| STS (COS. SPEA. ↑) | 0.2840 | 0.4569 | 0.3656 | 0.4331 | 0.4596 |
| BIOSSES | 0.2697 | **0.6363** | 0.4927 | **0.5891** | **0.5406** |
| SICK-R | **0.4981** | **0.5095** | 0.4482 | 0.4494 | **0.4612** |
| STS12 | 0.2307 | **0.3824** | 0.3017 | **0.3641** | **0.3905** |
| STS13 | 0.3603 | **0.5370** | 0.4292 | **0.4607** | **0.5755** |
| STS14 | 0.2045 | **0.4223** | 0.3550 | **0.4482** | **0.4980** |
| STS15 | 0.2068 | **0.5396** | 0.4513 | **0.5387** | **0.5489** |
| STS16 | **0.5413** | **0.5229** | 0.4721 | 0.5208 | **0.5635** |
| STS17 | 0.1230 | **0.2122** | -0.0283 | **0.1358** | **0.1456** |
| STS22 | 0.0922 | **0.4334** | 0.3632 | **0.4203** | **0.4313** |
| STSBenchmark | 0.3135 | **0.3742** | 0.3716 | **0.4045** | **0.4414** |
| RERANKING (MAP ↑) | 0.3758 | 0.4163 | 0.3704 | 0.4137 | 0.4104 |
| AskUbuntuDupQuestions | 0.4352 | **0.4774** | 0.4577 | **0.4767** | **0.4821** |
| MindSmallReranking† | **0.2817** | 0.2815 | 0.2815 | **0.2816** | **0.2818** |
| SciDocsRR | 0.5188 | **0.5682** | 0.4427 | **0.5744** | **0.5647** |
| StackOverflowDupQuestions | 0.2675 | **0.3383** | 0.2997 | **0.3221** | **0.3133** |
| PAIRCLASSIFICATION (AVG. PRE. ↑) | 0.2914 | 0.5316 | 0.5605 | 0.5547 | 0.5754 |
| SprintDuplicateQuestions | 0.0840 | 0.4954 | **0.5239** | **0.5686** | **0.5528** |
| TwitterSemEval2015 | 0.3846 | **0.4106** | **0.4151** | 0.3510 | **0.4221** |
| TwitterURLCorpus | 0.4058 | 0.6890 | **0.7425** | **0.7446** | **0.7513** |
| SUMMARIZATION (COS. SPEA. ↑) | 0.2042 | 0.1964 | 0.2470 | 0.2346 | **0.2774** |
| SummEval | 0.2042 | 0.1964 | **0.2470** | **0.2346** | **0.2774** |

Table 13: Performance comparison on the STS-B benchmark (Spearman correlation) between GLOT and prompting-based methods across Encoder and Decoder architectures.

| Method (Encoder Architectures) | STS-B |
|---|---|
| ***BERT (Encoder-only)*** | |
| PromptBERT | 70.60 |
| GLOT (Ours) | **83.85** |
| ***Mistral-7B (Decoder-only)*** | |
| PromptEOL | 75.77 |
| Pretended CoT | 76.66 |
| Knowledge Enhancement | 74.09 |
| GLOT (Ours) | **80.51** |

Table 14: Performance comparison between GLOT (frozen backbone) and fully fine-tuned contrastive baselines. Runtime is reported in milliseconds per batch. Best performance is in **bold**.

| Model | Runtime (ms) ↓ | Params (M) ↓ | CoLA (MCC) ↑ | STS-B (Spear.) ↑ | RTE (ACC) ↑ |
|---|---|---|---|---|---|
| SBERT (MPNet-v2) (Song et al., 2020) | 52.99 ± 0.08 | 109.50 | 17.70 | 86.70 | 54.51 |
| SimCSE (BERT-sup) (Gao et al., 2021) | 47.21 ± 0.09 | 109.50 | 24.91 | **87.27** | 52.70 |
| GLOT (BERT) | **13.40 ± 3.00** | **8.92** | **47.49** | 83.86 | **59.21** |

Table 15: Ablation study comparing GLOT against a parameter-matched MLP baseline using the Mistral-7B backbone. Best results are in **bold**.

| Method | Params (M) ↓ | CoLA (MCC) ↑ | STS-B (Spear.) ↑ | RTE (ACC) ↑ |
|---|---|---|---|---|
| MLP | 9.2 | 51.33 | 74.12 | 57.76 |
| GLOT (Ours) | **8.9** | **54.30** | **80.51** | **59.21** |

Table 16: Inference-time cost and performance benchmark using the Mistral-7B backbone. Runtime is measured in seconds per batch. Best performance is in **bold**.

| Method | # Params | GPU Mem (GB) ↓ | Runtime (s) ↓ | MCC ↑ | Spear. ↑ | ACC ↑ |
|---|---|---|---|---|---|---|
| [EOS] | 8.2K | 32.58 | 3.227 ± 0.006 | 38.63 | 72.36 | 50.90 |
| Mean | 8.2K | 32.58 | 3.244 ± 0.003 | 38.61 | 77.96 | 53.07 |
| Max | 8.2K | 32.58 | 3.259 ± 0.009 | 10.78 | 70.72 | 53.07 |
| AdaPool | 2.1M | 32.58 | 3.249 ± 0.011 | 48.00 | 79.55 | 54.87 |
| GLOT (Ours) | 8.92M | 32.64 | 3.252 ± 0.007 | **53.29** | **80.51** | **59.21** |

Table 17: Unified comparison of training efficiency and performance using the Mistral-7B backbone. We categorize methods by computational cost. Best results are in **bold**.

| Category | Method | Trainable Params | GPU Mem (GB) | CoLA (MCC) | STS-B (Spear) | RTE (Acc) |
|---|---|---|---|---|---|---|
| **Low Cost** | Mean Pooling | 0 | < 0.1 | 38.61 | 77.96 | 53.07 |
| | Max Pooling | 0 | < 0.1 | 10.78 | 70.72 | 53.07 |
| | AdaPool | 2.1 M | < 0.1 | 48.00 | 79.55 | 54.87 |
| | MLP Baseline | 9.2 M | 0.42 | 51.33 | 74.12 | 57.76 |
| **High Cost** | LoRA ($r = 64$) | 167.8 M | 33.5 | 48.23 | 54.54 | 53.43 |
| | Full Fine-Tuning | 7,110 M | > 40.0 | 49.63 | 55.68 | 55.23 |
| **Proposed** | GLOT (Ours) | **8.9 M** | **0.42** | **53.29** | **80.51** | **59.21** |

