# OpenReview forum: "Towards Improved Sentence Representations using Token Graphs"
_ICLR.cc/2026/Conference — ICLR 2026 Poster_

### Official Review · Reviewer_aPBo · 2025-10-29

**Soundness:** 2
**Presentation:** 2
**Contribution:** 2
**Rating:** 4
**Confidence:** 5

**Summary:**

This paper proposes GLOT which is a novel pooling methodology that builds graph out of latent representations and then pass through them multiple layers of GNN and lastly using the recent learnable pooling technique called AdaPool to capture sentence level representations. It operates on output tokens which comes from a frozen backbone so it has very low both computational and memory overhead. It shows significant improvements over static or learnable pooling across benchmarks like GLUE and other downstream tasks from MTEB, and IMDB long text classification dataset.

**Strengths:**

* The paper introduces an interesting approach to token pooling by using GNN-based graph construction to preserve token relationships that standard pooling methods discard.

* The proposed architecture is lightweight, requiring only ~9M trainable parameters and 0.42GB GPU memory compared to fine-tuning approaches.

* The method shows consistent improvements over frozen baseline pooling methods across multiple model architectures from BERT to Mistral-7B.

* The diagnostic stress test provides useful insights into robustness under noisy conditions, where the method maintains performance while baselines degrade.

* The paper includes reasonable ablation studies on graph sparsity and component contributions with sufficient implementation details for reproducibility.

* Operating on frozen backbones makes the approach practically accessible for resource-constrained settings without expensive fine-tuning infrastructure.

**Weaknesses:**

*  The evaluation approach may not fully align with the paper's focus on sentence representations, as it relies heavily on GLUE tasks rather than comprehensive MTEB evaluation, which is typically considered the standard benchmark for assessing sentence embedding quality.

* The MTEB results are limited to 7 out of 58 tasks, which makes it difficult to assess the method's performance across the full range of sentence embedding applications and may not fully represent its general capabilities.

* Comparisons to established sentence embedding methods like contrastive fine-tuning approaches (Sentence-BERT, SimCSE) or recent lightweight adaptation methods (LLM2Vec) would strengthen the evaluation and provide better context for the contributions.

* The baseline comparisons involve different parameter counts (~9M for GLOT vs. minimal parameters for simple pooling), which makes it somewhat unclear whether improvements stem primarily from the graph-based design or from having additional learnable capacity.

* The efficiency analysis in Table 5 focuses on CoLA, which emphasizes grammatical acceptability rather than semantic similarity, and the surprisingly strong performance relative to full fine-tuning raises questions about baseline tuning that could be addressed with more details.

* Information about inference-time computational costs, particularly for graph construction and GNN processing, would help readers better understand the practical deployment trade-offs of the approach.

* Including parameter-matched baselines such as deeper MLP-based pooling would help better isolate the specific contribution of the graph-based relational learning component.

* A unified comparison showing both efficiency and performance metrics for all methods in a single analysis would provide clearer insights into the overall trade-offs and help readers make more informed assessments.

**Questions:**

1. Could you provide complete MTEB results across all 58 tasks to give a comprehensive picture of the method's performance on standard sentence embedding benchmarks?

2. How does GLOT compare to established sentence embedding methods like contrastive fine-tuning (Sentence-BERT, SimCSE) and recent lightweight approaches (LLM2Vec) in terms of both performance and computational cost?

3. What is the performance of parameter-matched baselines, such as multi-layer MLP pooling with similar capacity (~9M parameters), to isolate whether gains come from graph-based relational learning or simply from having more learnable parameters?

4. Could you clarify the training procedures for the full fine-tuning and LoRA baselines in Table 5, given that GLOT unexpectedly outperforms these much larger capacity methods?

5. What are the inference-time computational costs including graph construction and GNN forward pass, and how do these compare to simpler pooling methods?

6. How does GLOT perform when the backbone is fine-tuned rather than frozen, and does it provide complementary benefits to fine-tuning or does fine-tuning alone achieve comparable results?

7. Could you provide a unified comparison table showing both efficiency metrics and performance across multiple representative tasks for all methods to enable clearer assessment of trade-offs?

---

> ### Author Response · Authors · 2025-11-21
> **Response (Part 1)**
>
> We thank the reviewer for their thorough evaluation and for recognizing GLOT’s “consistent improvements” and “lightweight” nature. We also appreciate your thorough and constructive feedback, which we address in our responses below. We hope that you find them satisfactory, and that you will consider revising your score.
>
>
> ---
>
> **Q1**. Results on MTEB
>
> **A1**. We thank you for the constructive feedback which we embrace. To address your comment, we have now significantly expanded our evaluation to 44 out of 56 MTEB tasks using the Mistral-7B backbone. The remaining evaluation datasets are significantly larger, with some containing 350M sentences which are not yet complete during the rebuttal timeframe. We are currently working to add them. We commit to including the 56-task suite in the final revision of our paper, and report them here as soon as we have the results. The results in the Table below, *with full results on all baselines in Table 12 of our revision, rendering the overall evaluation of 44 datasets and baselines* further highlight the competitive performance of GLOT, which is also state-of-the-art performance on sentence structure-dependent and retrieval tasks. We also followed your advice and reframed our positioning in the abstract. Concretely, from the results in Table 12 in the revised paper, we see that GLOT is consistently offering competitive performance with state-of-the-art techniques. It also outperforms the runner-up method (Mean Pooling) on Retrieval (+18.3% relative improvement), Summarization (+41.2% relative), and Pair Classification (+8.2% relative). These results further establish the effectiveness of our GLOT approach that explicitly modeling and learning token interactions improves performance, in addition to its ability to filter out distractor tokens, as shown in our Stress Test in Table 7.
>
> ---
>
> **Q2**. How does GLOT compare to finetuning based contrastive learning baselines like SBERT, SimCSE or LLM2Vec?
>
> **A2**.  Thank you for the suggested baselines which we welcome and now add comparisons with. The key distinction to note here is that SBERT, SimCSE, and LLM2Vec require updating the LLM (or altering attention mechanisms, as in LLM2Vec). In contrast, GLOT aggregates token embeddings by relational learning from token interactions of a frozen LLM. To illustrate this tradeoff, we compare below GLOT (with a BERT model) with fully fine-tuned SBERT (MPNet) and SimCSE (BERT-sup) on three diverse GLUE tasks (scores are multiplied by 100 and the best performance is in **bold**):
>
> | Model                                      | Runtime (ms) $\downarrow$       | Params (M) $\downarrow$ | CoLA (MCC $\uparrow$) | STSB (Spear. $\uparrow$) | RTE (ACC $\uparrow$) |
> |--------------------------------------------|----------------------|--------------|-----------|-----|-----|
> | SBER (MPNet-v2)        | 52.99 ± 0.08         | 109.50       | 17.70     | 86.70 | 54.51 |
> | SimCSE (BERT-sup)             | 47.21 ± 0.09         | 109.50       | 24.91     | **87.27** | 52.70 |
> | GLOT (BERT)                        | **13.40 ± 3.00**     | **8.92**     | **47.49** | 83.86 | **59.21** |
>
> As can be seen from the Table, our GLOT outperforms the fully fine-tuned baselines on CoLA (+22.5 points) and RTE (+6.5 points), which require complex linguistic understanding. While SimCSE is slightly specialized for semantic similarity (STS-B), GLOT remains highly competitive while also being efficient with $\approx 12\times$ fewer trainable parameters. We added this comparison in Table 14 in the revision. We believe that these additional experiments further highlight the effectiveness of GLOT. Thank you.

---

> > ### Author Response · Authors · 2025-11-21
> > **Response (Part 2)**
> >
> > **Q3**. How does GLOT compare to parameter-matched (approximately 9M) baseline like a deeper MLP?
> >
> > **A3**.  Thank you for the insightful question. To accommodate your suggestion, we replace Token-GNN with an MLP with a similar number of parameters, and measure the performance using Mistral-7B on three diverse tasks from the GLUE benchmark. We show the results below, and added them to Table 15 of Appendix D.6 in the revised paper:
> >
> > | Method | Params (M) $\downarrow$ | CoLA (MCC $\uparrow$) | STS-B (Spear. $\uparrow$) | RTE (ACC $\uparrow$) |
> > |:------:|:-----------------------:|:---------------------:|:-------------------------:|:--------------------:|
> > | MLP    | 9.2                     | 51.33                 | 74.12                     | 57.76                |
> > | GLOT   | **8.9**                     | **54.30**             | **80.51**                 | **59.21**            |
> >
> > These results indicate that GLOT consistently outperforms the parameter-matched (9M parameters) MLP baseline across all three tasks. The performance gap is especially pronounced on STS-B (+6.4 points), where modeling fine-grained semantic similarity is critical. This difference arises from a fundamental limitation of the MLP which processes the input as a set of independent token embeddings instead of explicitly modeling token-token relationships. In contrast, GLOT employs a GNN over the token graph, allowing tokens (nodes) to exchange information.  This ablation study confirms that the superior performance of GLOT when compared to other baselines comes from the relational learning approach in GLOT, further establishing its effectiveness.  Thank you for the important suggestion.
> >
> > ---
> >
> > **Q4**. Could you clarify the training procedures for full fine-tuning and LoRA baseline (in Table 5) and explain why GLOT outperforms these larger capacity methods?
> >
> > **A4**. Thank you for the question. Our training procedures closely follow the precedence set by Taori et al., 2023 and Li et al., 2024. On each dataset of CoLA, STSB and RTE datasets, we fine-tune the Mistral-7B on the training splits for 3 epochs, with a learning rate of $2\times 10^{-5}$ and weight decay of $0.01$. For LoRA, we use the rank hyperparameter $r = 64$ for the attention and feed-forward blocks with a learning rate $2 \times 10^{-4}$ and weight decay of $0.01$.  From Table 5, we see that GLOT outperforms full fine-tuning and LoRA counterparts on these datasets, and we attribute this to parameter-scale of backbone LLM, which is 7B for Mistral; Prior research (Li et al., 2024 and Saroufim et al., 2025) indicates that catastrophic forgetting becomes severe when the parameter scale of the fine-tuned LLM on downstream tasks increases. This, along with equivalence between LoRA and full fine-tuning (Shuttleworth et al., 2025) support our findings in Table 5. We included this important discussion about the training procedures in our Implementation Details (Appendix B) and elaborated on the discussion for Table 5 in our revision. Thank you for the thoughtful question.
> >
> > #### References
> >
> > Li et al., Revisiting Catastrophic Forgetting in Large Language Model Tuning, EMNLP 2024
> >
> > Saroufim et al., NeurIPS 2023 LLM Efficiency Fine-Tuning Competition, arXiv
> >
> > Shuttleworth et al., LoRA vs Full Fine-tuning: An Illusion of Equivalence, NeurIPS 2025
> >
> > Taori et al., Stanford alpaca: An instruction-following llama model. Stanford blog, 2023.

---

> > > ### Author Response · Authors · 2025-11-21
> > > **Response (Part 3)**
> > >
> > > **Q5**. What are the inference-time computational costs including graph construction and GNN forward pass, and how do these compare to simpler pooling methods?
> > >
> > > **A5**. Thank you for the question. Below, we benchmark the inference-time costs of GLOT (including graph construction and GNN forward pass) and other pooling methods using Mistral-7B backbone, along with performance on representative datasets from GLUE benchmark:
> > >
> > > | Method   | # Params | GPU Memory (GB) $\downarrow$ | Runtime (s) $\downarrow$ | MCC $\uparrow$ | Spear. $\uparrow$ | ACC $\uparrow$ |
> > > |:--------:|:--------:|:--------:|:-----:|:------:|:-----:|:-----:|
> > > |   [EOS]   |   8.2K | 32.58  | 3.227 $\pm$ 0.006 | 38.63 | 72.36 | 50.90 |
> > > | Mean | 8.2K | 32.58 | 3.244 $\pm$ 0.003 | 38.61 | 77.96 | 53.07 |
> > > | Max | 8.2K | 32.58 | 3.259 $\pm$ 0.009  | 10.78 | 70.72 | 53.07 |
> > > | Adapool | 2.1M | 32.58 | 3.249 $\pm$ 0.011 | 48.00 | 79.55 | 54.87 |
> > > | GLOT (ours) | 8.92M | 32.64 | 3.252 $\pm$ 0.007  | **53.29** | **80.51** | **59.21** |
> > >
> > > The inference time consists of forward pass through the LLM backbone and through the pooling module. Hence, inference-time costs are almost identical across the pooling methods, with GLOT requiring only 600 MB additional GPU memory and 3ms additional time when compared to baseline pooling methods (e.g., 32.64 GB and 3.252 ms for GLOT vs 32.58 GB and 3.249 ms for AdaPool). This is identical to the rest of the baselines due to specialized matrix multiplication and sparse computation operations from Pytorch and Pytorch Geometric thereby offering significant performance improvements for GLOT. Additionally, following the suggestions of **Reviewer 56SJ**, we conducted a graph construction runtime analysis, which we find to be relevant to your question as well. The results are reported in the Table below:
> > >
> > > |  Backbone  | Max Context Length    | Graph Construction (ms) | Total Runtime (ms) | Overhead |
> > > |:----------:|:---:|:------------------:|:--------------:|:---------------------:|
> > > | BERT / RoBERTa |  512  | 0.043 $\pm$ 0.002  | 5.36 $\pm$ 0.06 | 0.8% |
> > > | SmolLM2 | 8192 | 4.455  $\pm$   0.047 | 772.30 $\pm$ 0.11 | 0.5% |
> > > | TinyLlama - 1.1 B| 2048 | 0.672  $\pm$   0.001 | 143.15 $\pm$ 0.05 | 0.6% |
> > > | Llama - 3B| 8192 | 15.046 $\pm$ 0.373 | 2041.77 $\pm$ 0.19 | 0.7% |
> > > | Mistral-7B | 32768 |  303.466 $\pm$   1.024  | 23460.29 $\pm$ 4.77| 1.3% |
> > >
> > >
> > > These results further indicate that GLOT offers an efficient and effective token pooling method for frozen LLMs by learning from token graphs. We included this discussion in Tables 16 and 10 of our revision. Thank you for this important suggestion.
> > >
> > > ---
> > >
> > > **Q6**. Does GLOT provide complimentary benefits when the backbone is finetuned instead of frozen or does fine-tuning alone achieve comparable results?
> > >
> > > **A6**. This is a very interesting question. GLOT is technically compatible with fine-tuned backbones. However, our results (Table 1 & 5) show that when the backbone is frozen, GLOT offers similar or better performance compared with fully fine-tuned models (e.g., matching Full FT performance on CoLA), achieving our primary goal of effective utilization of LLM embedding accompanied with efficiency without the high costs of LLM weight updates. This work presents the concept of learning over token graphs, and we believe a fine-tuned LLM coupled with GLOT setting is an exciting future research direction that warrants its own study. We revised the paper to include this discussion in Section B.2. Thank you.

---

> > > > ### Author Response · Authors · 2025-11-21
> > > > **Response (Part 4)**
> > > >
> > > > **Q7**. Could you provide a unified table summarizing results across all the experiments?
> > > >
> > > > **A7**.  Thank you for the question. Following your guidance, we provide a unified comparison using the Mistral-7B backbone. We consider efficiency and performance on three diverse tasks: CoLA, STS-B, and RTE.
> > > >
> > > >
> > > > | Category | Method | Trainable Params | GPU Mem (GB) | CoLA (MCC) | STS-B (Spear) | RTE (Acc) |
> > > > | :--- | :--- | :--- | :--- | :--- | :--- | :--- |
> > > > | **Low Cost** | Mean Pooling | 0 | < 0.1 | 38.61 | 77.96 | 53.07 |
> > > > | | Max Pooling | 0 | < 0.1 | 10.78 | 70.72 | 53.07 |
> > > > | | AdaPool | 2.1 M | < 0.1 | 48.00 | 79.55 | 54.87 |
> > > > | | **MLP** | 9.2 M | 0.42 | 51.33 | 74.12 | 57.76 |
> > > > | **Higher Cost** | LoRA (r=64) | 167.8 M | 33.5 | 48.23 | 54.54 | 53.43 |
> > > > | | Full Fine-Tuning | 7,110 M | > 40.0 | 49.63 | 55.68 | 55.23 |
> > > > | **Proposed** | GLOT (Ours) | **8.9 M** | **0.42** | **53.29** | **80.51** | **59.21** |
> > > >
> > > > When compared to baselines, GLOT introduces a minor parameter increase over AdaPool (8.9M vs 2.1M) but yields significant gains across all metrics (e.g., +5.3 points on CoLA and +4.3 points on RTE). If we consider a parameter count matched MLP instead of our GLOT, we still see a performance gap in the favor of our GLOT. When compared to  fine-tuning, GLOT outperforms LoRA and full fine-tuning on all three tasks while requiring ~19x fewer parameters than LoRA and ~800x fewer than full fine-tuning, with a fraction of the memory usage.
> > > >
> > > > This summary of our results further highlights and confirms that GLOT offers a "sweet spot" approach: it offers a performance that is competitive or better with fine-tuning techniques, and maintains the computational efficiency of frozen methods. We added this Table to the revised paper in Table 17 and Section D.8. Thank you for the suggestion that in our opinion helps to express the contribution and merit of GLOT.
> > > >
> > > > ---
> > > >
> > > > ### Summary
> > > >
> > > > We appreciate your constructive feedback that helped us to improve the quality of the paper, in our opinion. We remain committed to addressing any additional questions you may have, and hope that you will consider revising your score.  Thank you.

---

> > > > > ### Comment · Reviewer_aPBo · 2025-11-25
> > > > >
> > > > > * I appreciate authors colloboration to adress paper's shortcomings. I believe MTEB result is a must and I do understand cost of eval in MTEB but I am fine with full results being added after the rebuttal.
> > > > >
> > > > > * An interesting analysis would be how do the contribution of tokens change compared to something like mean pool where they attend uniformly which is probably suboptimal.
> > > > >
> > > > >
> > > > > * Finally, I raised my score to 6.

---

> > > > > > ### Author Response · Authors · 2025-12-03
> > > > > >
> > > > > > Dear Reviewer aPBo,
> > > > > >
> > > > > > We are sincerely grateful for your continued engagement and for raising your score to a 6. We appreciate your recognition of our efforts to address your concerns.
> > > > > > We also welcome your suggestion, and to accommodate it, we have conducted an analysis of Token Contributions. We have added a new section (Appendix D.9 in the revised paper) and **Figure 5** to the manuscript, and we discuss it in more detail, below. Thank you.
> > > > > >
> > > > > > ---
> > > > > >
> > > > > > 1. **Analysis of Token Contributions:** To understand how GLOT compares to uniform weighting (Mean Pooling) and other learnable pooling methods (AdaPool), we visualized the token contribution weights ($\pi$) on samples from the Quora Question Pairs dataset using a frozen BERT backbone.
> > > > > > Our analysis highlights distinct differences in how these methods prioritize information:
> > > > > >  - Mean Pooling (Uniform): As you hypothesized, Mean Pooling assigns a uniform weight ($1/L$) to all tokens. This causes signal dilution, where tokens with little semantic value (e.g., “is”, “of”, “?”) contribute equally to the final representation as the critical semantic tokens.
> > > > > > - AdaPool (Independent): While non-uniform, AdaPool treats tokens independently. We observed that it frequently overfits to common syntactic markers or functional words. For example, in the query "A person is riding a horse", AdaPool places high weight on the article "a" and the noun "person".
> > > > > > - GLOT (Ours): GLOT exhibits a highly selective distribution. By learning token interactions with a GNN, GLOT successfully identifies **semantically important tokens**.
> > > > > >     - In the query "What is the maximum number of genetically unique individuals that human genome allows?", GLOT suppresses the interrogative "What" and assigns maximum weight to **"genome"** and **"individuals"**.
> > > > > >     - In the query "A person is riding a horse", GLOT isolates the specific action **"riding"** and the object **"horse"**.
> > > > > >
> > > > > > This analysis confirms that our GLOT improvements stem from its ability to use graph structures to prioritize semantic content and filter out the distractors inherent in natural language. Thank you for the thoughtful suggestion.
> > > > > >
> > > > > > 2. **Update on MTEB Experiments**: Regarding the MTEB evaluation, we are happy to report that we have now completed over 250 experiments since our last update. The results from these additional runs consistently align with our previous findings, and the claims stated in our initial response continue to hold and they are included in the current revision. We remain fully committed to including the complete suite of results (remaining 6/56 tasks, with dataset sizes in the 5M-300M range) in the final camera-ready version. Thank you.
> > > > > >
> > > > > > ---
> > > > > >
> > > > > >
> > > > > > We thank you again for your valuable feedback, which we find to have significantly strengthened our evaluation and analysis, and the overall quality of the paper.
> > > > > >
> > > > > > Thank you, and best regards, \
> > > > > > Authors.

---

### Official Review · Reviewer_gK63 · 2025-10-29

**Soundness:** 3
**Presentation:** 3
**Contribution:** 3
**Rating:** 6
**Confidence:** 4

**Summary:**

This paper presents GLOT, a new method for deriving sentence embeddings. Compared to naive mean or max pooling, GLOT explicitly models the semantic relationships between tokens by incorporating a Graph Neural Network (GNN) structure before the final aggregation step. Experimental results demonstrate that GLOT can achieve effective performance improvements with minimal computational resources while keeping the backbone model frozen.

**Strengths:**

1. Sentence embedding is a core topic in representation learning, and pooling is a crucial step in converting token-level embeddings into sequence-level embeddings. Therefore, the chosen track is highly relevant to the conference's scope and holds significant practical value.
2. The paper is well-written. The methodology is presented clearly and intuitively. The authors claim that GLOT is the first work to learn sentence representations via GNNs on top of a frozen Large Language Model (LLM).
3. The experiments are conducted on both discriminative PLMs (BERT, RoBERTa) and generative PLMs (Llama, Mistral), validating the method's effectiveness across different model architectures.

**Weaknesses:**

1. The MTEB is a massive benchmark for embedding evaluation. The authors show that their method surpasses baselines on only a selected subset of tasks. This seems to be insufficient to support the claim of "state-of-the-art performance" in the abstract.
2. The chosen baselines are rather conventional. I am curious about the comparison between this proposed method and the prompting-based methods that have emerged in the past two years. Prompting methods can have an even smaller memory footprint than this work, as they require no parameter updates at all. Some relevant references include:
-	“PromptBERT: Improving BERT Sentence Embeddings with Prompts”
-	“Scaling Sentence Embeddings with Large Language Models”
-	“Simple Techniques for Enhancing Sentence Embeddings in Generative Language Models”
3. The authors construct the graph based on the similarity between token embeddings. This approach is relatively straightforward for discriminative PLMs with bidirectional attention. However, for generative PLMs with unidirectional attention, tokens that appear later in a sequence have a larger attention scope. Shouldn't the similarity calculation between their token embeddings account for this disparity?

**Questions:**

Authors should give their response and explanation with respect to the weaknesses mentioned above.

---

> ### Author Response · Authors · 2025-11-21
> **Response (Part 1)**
>
> We thank the reviewer for the overall positive evaluation of our work. We are encouraged that you found the paper “well-written” and the methodology “clear and intuitive”. We particularly appreciate your recognition that GLOT is the **first work** to learn sentence representations via GNNs on frozen LLMs and that the approach holds “significant practical value” for resource-constrained settings. We are also thankful for the constructive feedback. Below,  we provide our responses to each of your comments.
>
> ---
>
> 1. **Evaluation on MTEB:** We thank you for the constructive feedback which we embrace. To address your comment, we have now significantly expanded our evaluation to 44 out of 56 MTEB tasks using the Mistral-7B backbone. The remaining evaluation datasets are significantly larger, with some containing 350M sentences which are not yet complete during the rebuttal timeframe. We are currently working to add them. We commit to including the 56-task suite in the final revision of our paper, and report them here as soon as we have the results. The results in the Table below, *with full results on all baselines in Table 12 of our revision, rendering the overall evaluation of 220 experiments (44 datasets, 5 baselines)* further highlight the competitive performance of GLOT, which is also state-of-the-art performance on sentence structure-dependent and retrieval tasks. We also followed your advice and reframed our positioning in the abstract. Concretely, from the results in Table 12 in the revised paper, we see that GLOT is consistently offering competitive performance with state-of-the-art techniques. It also outperforms the runner-up method (Mean Pooling) on Retrieval (+18.3% relative improvement), Summarization (+41.2% relative), and Pair Classification (+8.2% relative).  These results further establish the effectiveness of our GLOT approach that explicitly modeling and learning token interactions improves performance, in addition to its ability to filter out distractor tokens, as shown in our Stress Test in Table 7.
>
> ---
>
> 2. **Clarification on the choice of baselines and Comparison to Prompt-Based methods:** Thank you for the suggestion for additional baselines which we embrace. Our goal in this work is to treat token embeddings for a given sentence from an LLMs as a semantic graph in the latent space, and reframe pooling as relational learning from token interactions instead of treating them as a set of independent vectors like in existing baselines shown in our paper. *Orthogonally*, prompting uses hand-crafted prefixes (which are the pre-defined across all sentences and datasets) to obtain a semantically rich representation for the sentence. Differently, our GLOT approach suggests learning over token-graphs, which is a different and unique approach. It is important to note that the computational efficiency of GLOT is achieved because our approach does not require storing gradients of the LLM in memory, as well as amortizing its forward pass, to obtain token embeddings, as a dataset preparation step. Nonetheless, we agree that it is beneficial to compare our approach with prompt-based techniques, and below we compare GLOT with your proposed baselines: PromptBERT(Jiang et al., 2022), PromptEOL (Jiang et al., 2024), Pretrained CoT and Knowledge Enhancement (Zhang et al., 2024) on the STS-B benchmark. The results, reported in the Table below, show thatGLOT consistently outperforms prompting methods on both encoder and decoder architectures.
>
> | Method | STSB (Spearman) |
> |:------:|:---------------:|
> |  PromptBERT  |  70.60    |
> | GLOT BERT | **83.85** |
>
> | Method | STSB (Spearman) |
> |:------:|:---------------:|
> | PromptEOL Mistral -7B | 75.77 |
> |  Pretended CoT Mistral-7B | 76.66 |
> | Knowledge Enhancement Mistral-7B| 74.09 |
> | GLOT Mistral-7B | **80.51** |
>
> These results confirm that relational learning over token-graphs is an effective approach. We added this discussion and these results in Table 14 and Section D.4 of our revision. Thank you.
>
> #### References
>
> Jiang et al., PromptBERT: Improving BERT Sentence Embeddings with Prompts, EMNLP 2022
>
> Jiang et al., Scaling Sentence Embeddings with Large Language Models, EMNLP 2024
>
> Zhang et al., Simple Techniques for Enhancing Sentence Embeddings in Generative Language Models, ICIC 2024

---

> > ### Author Response · Authors · 2025-11-21
> > **Response (Part 2)**
> >
> > 3. **Influence of uni-directional attention in generative PLMs on token similarity:** We appreciate your insightful comment. This is an interesting point. Recent research studies attention sinks (Xiao et al., 2024, Gu et al., 2025) and position bias (Wu et al., 2025) in autoregressive LLMs, where the attention scores are larger for earlier token positions, and mitigate this problem by fine-tuning the backbone with bidirectional attention. Differently, when obtaining sentence representations, the LLM has access to all the tokens at inference time, and GLOT utilizes them by creating a cosine similarity based graph allowing the GNN to propagate information from later to earlier tokens. This explains why GLOT yields significant gains on decoders (e.g., +13.13% on CoLA for Mistral, Table 1). Furthermore, our graph-based formulation opens the door to studying additional token interaction modeling techniques as future research work. For instance, recent work on "graph rewiring" (Barbero et al., 2024; Arnaiz-Rodriguez et al., 2022) and "virtual nodes" (Qian et al., 2024) have shown techniques in learning graph connectivity for graph learning tasks in GNNs. While GLOT is focused on introducing the concept of learning on token graphs already strong performance with cosine similarity, we view these dynamic rewiring strategies as an exciting avenue to further enhance the model's ability to capture long-range dependencies in future research works. We have added this discussion to our revised Conclusion. Thank you.
> >
> >
> > #### References
> >
> > Xiao et al., Efficient Streaming Language Models with Attention Sinks, ICLR 2024
> >
> > Gu et al., When Attention Sink Emerges in Language Models: An Empirical View, ICLR 2025
> >
> > Wu et al., On the Emergence of Position Bias in Transformers, ICML 2025
> >
> > Barbero et al., Locality Aware Graph Rewiring in GNNs, ICLR 2024
> >
> > Arnaiz-Rodriguez et al., DiffWire: Inductive Graph Rewiring via the Lovász Bound, LoG 2022
> >
> > Qian et al., Probabilistic Graph Rewiring via Virtual Nodes, NeurIPS 2024
> >
> > ---
> >
> > ### Summary
> >
> > We appreciate your constructive feedback that helped us to improve the quality of the paper, in our opinion. We remain committed to addressing any additional questions you may have, and hope that you will consider our responses in your final assessment.  Thank you.

---

### Official Review · Reviewer_BcGt · 2025-10-30

**Soundness:** 3
**Presentation:** 3
**Contribution:** 2
**Rating:** 4
**Confidence:** 4

**Summary:**

The paper proposes a lightweight structure-aware pooling module that reframes pooling as relational learning followed by aggregation, dubbed GLOT. The method is remarkably robust and efficient, as demonstrated by extensive experiments.

**Strengths:**

1) The paper has a clear motivation.
2) The method sounds technical.
3) GLOT has been evaluated through extensive experiments.

**Weaknesses:**

1) Some technical details of the method need to be presented.
2) The paper lacks essential theoretical explanations to ensure the effectiveness of the method.
3) The experimental analysis needs more powerful explanations.

I have significant doubts about the rationale behind the design of the proposed GLOT method and about why using a GNN-based approach alone can substantially improve the problem. These issues should be clearly explained in the method description, theoretical analysis, and experimental validation, but I have not found such clarification in the current version. I consider this to be the core of the paper.

**Questions:**

1) GLOT seems to only use an additional GNN to refine the representation. Why does it achieve such significant improvements in the results?
2) GLOT is easy to implement, and its effectiveness seems to come entirely from the GNN. Would using more advanced variants of GNN further improve the robustness of the model?
3) GLOT obviously has a more complex graph-based structure. What explains its lower number of parameters and faster training process?

---

> ### Author Response · Authors · 2025-11-21
> **Response (Part 1)**
>
> We thank you for noting the paper’s “clear motivation” and “extensive experiments”. We are also grateful for the constructive feedback that in our opinion is important for improving the quality of our paper. We provide our responses to each of your comments below. We hope that you find them satisfactory, and that you will consider revising your score.
>
> A common thread that we identify in your review is the rationale for why GLOT is effective, and we appreciate this opportunity to clarify. The questions in the review center on a single conceptual point that we will make more explicit: GLOT's performance stems from a fundamental *shift in the pooling paradigm itself*. Standard methods treat an LLM's token outputs as an independent set, which discards the rich relational structure between tokens and leads to the symptom of signal dilution. Our core approach in GLOT is that these tokens should be treated as a graph to model the interactions between token embeddings, and GLOT is an effective and efficient mechanism to learn from this structure. This shift from 'set-based' to 'graph-based' pooling explains the performance gains (Q1), the choice of a GNN (Q2), and the high efficiency (Q3). We elaborate and address your specific questions below. Thank you.
>
> ---
>
> **Q1**. Why does adding a GNN achieve such significant improvements?
>
> **A1**. Thank you for this question. The significant performance improvement acknowledged in your review, stems from a conceptual shift in how we frame the pooling mechanism itself. Standard pooling techniques (Mean, Max, and even learnable methods like AdaPool) may not be sufficient because it treats the LLM's token outputs as an *independent set of vectors*. This assumption has three key limitations. *First*, tokens are not independent. Linguistic theory (Firth, 1957) and recent analysis of Transformer geometry (Ethayaraj, 2019) confirm that token semantics are not token intrinsic, but arise from pairwise token dependencies (Clark et al., 2019 and Tenney et al., 2019). *Second*, while the LLM’s self-attention does capture the token-token relationships, it is typically trained for an objective like next-token prediction, which is not clearly aligned with other tasks like sentence-level classification. *Thirdly*, when a *set-based* pooling method is applied, this rich, task-relevant relational structure is discarded, leading to the symptom of signal dilution. A critical phrase like "not... good" can be averaged away by distractor tokens (please see more details in Figure 3 and Table 7).
> Different from set-based approach, our GLOT reframes the approach into a relational-based learning task. Inspired by the discussion above, we conjecture that the token hidden states are not only a set, but rather a graph in which nodes represent tokens and edges represent these dependencies, which can be further processed by the GNN in GLOT. It is therefore designed to perform relational learning by aggregating information over neighboring token embeddings before the final pooling . This allows it to learn complex, multi-token dependencies relevant to the task. This paradigm shift is significant because we are to the best of our knowledge the first to adapt the LLM's rich, yet possibly unoptimized, respect to sentence-level tasks, relational structure using a token-graph approach.  We appreciate the important comment, and we added this discussion to the revised paper in Section 3.2. Thank you.
>
> #### References
>
> Firth, J.R. (1957). A synopsis of linguistic theory 1930-1955.
>
> Clark et al, What Does BERT Look At? An Analysis of BERT's Attention., 2019 ACL Workshop
>
> BlackboxNLP: Analyzing and Interpreting Neural Networks for NLP
>
> Tenney et al., BERT Rediscovers the Classical NLP Pipeline., ACL 2019
>
> ---
>
> **Q2**. Could you provide a theoretical explanation for the effectiveness of GLOT?
>
> **A2**. We thank you for this question and allowing us to clarify the effectiveness of GLOT as a token pooling mechanism theoretically. The explanation stems from the fundamental difference between set-based and graph-based functions, which is discussed in our paper. As we state in "Properties of GLOT" (Section 3.2), standard pooling methods (Mean, Max, and AdaPool) fit to  the DeepSets (Zaheer et al., 2017) framework. The established theory for DeepSets and graph neural networks shows they are less powerful  than message-passing GNNs (Bronstein et al., 2021) as used in our GLOT . We also note that, a DeepSet is a special case of a GNN with no edge connectivity. Therefore, the graph-based approach in GLOT is theoretically more expressive. It can model linguistic phenomena that hinge on dependencies among the tokens, which set-based methods are theoretically blind to. This is the formal justification for GLOT's effectiveness. Following your question, we revised Section 3.2 to make this theoretical link more prominent.Thank you.

---

> > ### Author Response · Authors · 2025-11-21
> > **Response (Part 2)**
> >
> > **Q3**. Would using more advanced GNNs further improve robustness?
> >
> > **A3**. This is an excellent suggestion, which we embrace. We suggest that the main performance gain comes from the paradigm shift, i.e., from set-pooling to graph-pooling, rather than a specific GNN. Nonetheless, we followed your suggestion, and we ran new experiments with different, common GNN architectures: a Graph Convolutional Network (GCN) (Kipf et al., 2017) and a Graph Isomorphism Network (GIN) (Xu et al., 2019). For comparison, we also include the AdaPool baseline, which is a DeepSets-based approach. The results, reported in the Table below, show two takeaways:
> >
> > - The GLOT paradigm is robust with respect to the choice of the GNN: All graph-based methods (GAT, GCN, GIN) significantly outperform the set-based baseline (AdaPool), confirming our central claim that relational learning is the key.
> >
> > - Your intuition is correct: more advanced GNNs can further improve results. Our results indicate that GIN model outperforms GAT on CoLA (59.30 vs. 54.30) and RTE (59.30 vs. 59.21) for Mistral-7B, whereas for BERT, GAT is superior.
> >
> >
> >
> >
> > #### Mistral-7B
> > | Method                | CoLA (MCC) ↑ | STSB (Spearman) ↑ | RTE (ACC) ↑ |
> > |-----------------------|--------------|--------------------|-------------|
> > | AdaPool (No graph)  | 48.00        | 79.55              | 54.87       |
> > | GLOT (GCN)        | 52.65        | 79.74              | 57.04       |
> > | GLOT (GAT) (as in paper) | 54.30        | **80.51**              | 59.21       |
> > | GLOT (GIN)   | **59.30**        | 79.73              | **59.30**       |
> >
> >
> > #### BERT
> > | Method                | CoLA (MCC) ↑ | STSB (Spearman) ↑ | RTE (ACC) ↑ |
> > |-----------------------|--------------|--------------------|-------------|
> > | AdaPool (No graph)   | 29.20        | 80.01              | 51.62       |
> > | GLOT (GCN)  | 45.19        | 80.17              | 58.12       |
> > | GLOT (GAT) (as in paper) | 47.49        | **83.86**              | **59.21**       |
> > | GLOT (GIN)   | **47.78**        | 80.71        | 57.04  |
> >
> > This confirms our "set vs. graph" approach and establishes it as an efficient and novel approach for improving performance for sentence-level tasks from token embeddings. Beyond that, studying more GNN architectures in GLOT is an interesting future research direction. We have added this discussion and results in Appendix D3, Table 11 of our revised paper. Thank you.
> >
> > ##### References
> >
> > Kipf et al., Semi-Supervised Classification with Graph Convolutional Networks, ICLR 2017.
> >
> > Xu et al., How Powerful Are Graph Neural Networks, ICLR 2019

---

> > > ### Author Response · Authors · 2025-11-21
> > > **Response (Part 3)**
> > >
> > > **Q4**. How can GLOT be more complex (graph-based) but have fewer parameters and a faster training process?
> > >
> > > **A4**. Thank you for the thoughtful question. We appreciate this opportunity to clarify the source of GLOT’s effectiveness. The “complexity” of the graph structure does not translate to parameter count or training overhead for three key reasons:  (1) Non-Parametric Graph Construction: The graph topology (edges) is not learned via weights. Instead, it is derived dynamically from the data itself (the token hidden states) via cosine similarity. Thus, the complex connectivity adds zero learnable parameters;  (2) Parameter Sharing in GNNs: A defining feature of GNNs is that they apply the same set of learnable weights (the message-passing layer) to every node’s local neighborhood, regardless of the graph size (Bronstein et al., 2021), exploiting graph connectivity as an inductive bias (Battaglia et al., 2018 and Hamilton et al., 2017).  This allows GLOT to model complex, local and global dependencies with a fixed, compact set of weights ($\approx 8.9$M), compared to LoRA ($\approx 167.9$M) and full fine-tuning ($\approx 7$B) in Mistral; and (3) The design of GLOT: The efficiency gains in Table 5 compare GLOT with fine-tuning techniques. In those methods, the backbone parameters must be updated. In GLOT, by design, the multi-billion parameter backbone (e.g., Mistral 7B) is frozen i.e., there is no need to compute and store the gradients for the backbone in memory. This combination of a frozen backbone and a parameter-efficient GNN architecture yields the $> 100\times$ speedup.
> > >
> > > In addition, we also provide, in the Tables below, results from our response to **A2, A4 of Reviewer 56SJ** where we conduct additional studies using BERT analysing the performance and efficiency of GLOT, and the runtime for graph construction. As we can see, the memory and runtimes are 10x smaller for BERT. These results show that GLOT offers efficiency on a diverse set of LLM models, further strengthening the understanding of the efficiency and positioning of GLOT. Thank you.
> > >
> > > | Method   | No. Trainable Params | GPU Memory (GB) $\downarrow$ | Batch Runtime (ms) $\downarrow$ | MCC $\uparrow$ | Spear. $\uparrow$ | ACC $\uparrow$ |
> > > | :--- | :---: | :---: | :---: | :---: | :---: | :---: |
> > > | **Mistral-7B (Decoder)** | | | | | | |
> > > |   Full FT + [EOS]   |   7.11B | 32.59 | 1318.8 $\pm$ 1.1 | 49.63 | 55.68 | 55.23 |
> > > | LoRA (r = 64) + [EOS] | 167.8M | 33.50 | 1454.6 $\pm$ 1.1 | 48.23 | 54.54 | 53.43 |
> > > | GLOT (ours) | **8.92M** | **0.42** | **13.4 $\pm$ 3.0** | **53.29** | **80.51** | **59.21** |
> > > | **BERT-Base (Encoder)** | | | | | | |
> > > |   Full FT + [CLS]   |   109.5M | 0.74 | 52.5 $\pm$ 0.1 | 38.31 | 60.10 | 56.68 |
> > > | LoRA (r = 64) + [CLS] | 10.7M | 0.86 | 77.4 $\pm$ 0.2 | 36.16 | 64.64 | 56.32 |
> > > | GLOT (ours) | **8.92M** | **0.42** | **13.4 $\pm$ 3.0** | **47.49**  | **83.86** | **59.21** |
> > >
> > > |  Backbone  | Max Context Length    | Graph Construction (ms) | Total Runtime (ms) | Overhead |
> > > |:----------:|:---:|:------------------:|:--------------:|:---------------------:|
> > > | BERT / RoBERTa |  512  | 0.043 $\pm$ 0.002  | 5.36 $\pm$ 0.06 | 0.8% |
> > > | SmolLM2 | 8192 | 4.455  $\pm$   0.047 | 772.30 $\pm$ 0.11 | 0.5% |
> > > | TinyLlama - 1.1 B| 2048 | 0.672  $\pm$   0.001 | 143.15 $\pm$ 0.05 | 0.6% |
> > > | Llama - 3B| 8192 | 15.046 $\pm$ 0.373 | 2041.77 $\pm$ 0.19 | 0.7% |
> > > | Mistral-7B | 32768 |  303.466 $\pm$   1.024  | 23460.29 $\pm$ 4.77| 1.3% |
> > >
> > > #### References
> > >
> > > Battaglia et al. Relational inductive biases, deep learning, and graph networks. arXiv 2018
> > >
> > > Hamilton et al., Inductive Representation Learning on Large Graphs, NeurIPS 2017
> > >
> > > Bronstein et al., Geometric Deep Learning: Grids, Groups, Graphs, Geodesics, and Gauges., arXiv 2021
> > >
> > > ---
> > >
> > > ### Summary
> > > Our responses include added experiments, discussions, and clarifications to resolve your comments. We feel that your feedback helped us to improve our paper.  We have revised our manuscript accordingly, and our changes are marked in blue.  We hope that you find our responses satisfactory, and that you will consider revising your score. Thank you.

---

### Official Review · Reviewer_56SJ · 2025-11-01

**Soundness:** 2
**Presentation:** 3
**Contribution:** 3
**Rating:** 6
**Confidence:** 4

**Summary:**

This paper proposes a lightweight pooling module, called GLOT, which transforms token-level outputs of frozen LLMs into sentence embeddings through graph-based relational learning. Instead of treating tokens independently, GLOT builds a token-similarity graph, refines representations using a graph neural network, and aggregates them via a learnable readout. Across GLUE, IMDB, and MTEB, GLOT outperforms standard and learnable pooling methods while being 20× more parameter-efficient and 100× faster than fine-tuning approaches. It also shows strong robustness to noise, maintaining over 97% accuracy with 90% distractor tokens.

**Strengths:**

1. Reframing sentence pooling as token-relation modeling.
The paper introduces GLOT, a new framework that replaces traditional pooling (e.g., mean or [CLS]) with a graph-based aggregation process. GLOT unifies prior pooling schemes as special cases (when the graph is empty or uniform) and explicitly addresses the long-standing signal dilution problem in sentence embeddings. Demonstrated by strong improvements in the signal dilution test (≈97% accuracy under 90% distractors; Table 7, Sec. 5.4).

2. Strong consistency across diverse settings.
GLOT shows consistent, significant improvements over both static and learnable pooling baselines on major benchmarks: GLUE, IMDB, and MTEB. It outperforms methods such as mean pooling, weighted pooling, [CLS] pooling, across both encoder-based and decoder-based LLMs. This shows the method’s robust generalization and practical applicability across models and tasks. (Sec. 5.1–5.3, Tables 2–4, Fig. 3.)

3. High efficiency with minimal training cost.
The proposed module is extremely lightweight, only ≈ 8.9 M trainable parameters and ~ 0.42 GB GPU memory, compared to full fine-tuning or LoRA. Despite this small footprint, it delivers comparable or better accuracy and over 100× training-speed improvement. This efficiency makes GLOT highly practical for large-scale or resource-constrained deployments, supporting the paper’s claim of being a drop-in, low-cost alternative to fine-tuning. (Sec. 5.5, Table 5.)

**Weaknesses:**

1. While GLOT introduces a graph-based pooling paradigm, the graph construction step is heuristic; edges are formed by thresholding pairwise cosine similarities between token embeddings.
The paper does not explore learnable or adaptive graph formation, nor analyze how different thresholds quantitatively affect representation quality beyond a small ablation (τ = 0.4–0.6 works best). As a result, the approach lacks a deeper theoretical justification on how graph topology influences performance. (Sec. 3.2, Sec. 5.5, Table 6.)

2. Although Table 5 reports large efficiency gains (≈100× faster, 0.42 GB GPU), this measurement is only for one dataset (CoLA) and one backbone (Mistral-7B). The paper does not provide cross-task or cross-model runtime, throughput, or latency benchmarks, and omits the cost of graph construction itself. Therefore, the claimed computational advantages, while promising, are not fully validated across broader conditions.

3. The related-work section briefly mentions prior pooling and token-interaction models but does not clearly position GLOT relative to recent work. This leaves ambiguity about whether the proposed approach is a substantial conceptual advance or an effective adaptation of known ideas.

4. The first step of GLOT involves computing a pairwise cosine similarity matrix between all L tokens in the sequence to determine the edges. This is an O(L^2) operation with respect to the sequence length L. While the paper tests on sequences up to 512 tokens (IMDB)9, this O(L^2) step could become a significant computational bottleneck for applications involving very long contexts (e.g., thousands of tokens), a limitation which is not discussed.

**Questions:**

see Weaknesses

---

> ### Author Response · Authors · 2025-11-21
> **Response (Part 1)**
>
> We sincerely thank the reviewer for the constructive feedback and for highlighting GLOT’s efficiency and robustness. We are encouraged that you find our method "highly practical" and "consistent." We are also grateful for your constructive and thoughtful feedback. Our responses to your specific comments are detailed below.
>
> ---
>
> **Q1** How does the graph topology influence the model performance? What are other choices for token-graph formation?
>
> **A1**.  Thank you for the question. Our goal in this work is to introduce the concept of GLOT, where we learn over token-graphs, offering a new perspective for sentence-level tasks. Our results indicate that the optimal graph sparsity is determined by the nature of the downstream task. As shown in the Tables below, structural tasks requiring precise dependency (Hewitt & Manning, 2019) modeling (e.g., CoLA (Warstadt et al., 2019), RTE (Bowman et al., 2016)) peak at higher sparsity thresholds ($\tau=0.4-0.6$). In these cases, the threshold is higher and hence more selective, favoring stronger similarities in the graph.  Moreover, we find that semantic similarity tasks (e.g., STS-B) benefit from denser graphs ($\tau=0.0-0.2$), as they rely on broader token associations rather than more specified connections.
>
> To analyze this behavior further, and inspired by your comment , we have expanded the experimented in our original submission (reported in Table 8) to provide a deeper, quantitative analysis of how graph connectivity influences performance by conducting two new ablations below. We *(1)* evaluated the influence of graph structure ($\tau$) on both encoder (BERT) and decoder (Mistral-7B) models; and *(2)* quantified the linear separability of the pooled representations.
>
> Specifically, to measure "representation quality," we performed a *linear probing experiment*. We used the CoLA dataset, a binary classification task. For each $\tau$, we first computed frozen sentence embeddings $z$ from GLOT (with no classifier head trained). We then trained a standard linear SVM classifier on these representations. The resulting accuracy ("Linear Probing" column) is a direct, quantitative measure of the linear separability of the pooled embeddings. The results are provided in the Tables below. We also added these results in Table 8 and the discussion in Appendix D.1 of our revision. While in this paper we focus on introducing the concept of GLOT, we also mention that studying additional approaches for graph connectivity (Barbero et al., 2024, Arnaiz-Rodriguez et al., 2022, Qian et al., 2024), and their influence on the quality of pooled representation GNNs is an interesting future direction. We have included this discussion as a potential future work in our revised Conclusion section. Thank you.
>
> | Model | STSB (Spear. $\uparrow$)  | CoLA (MCC $\uparrow$) | RTE (ACC $\uparrow$) | Linear Probing (CoLA, ACC $\uparrow)$ |
> | :---: | :---: | :---: | :---: | :---: |
> | BERT ($\tau = 0.0$) | 81.88 | 39.62 | 50.90 | 76.41 |
> | BERT ($\tau = 0.2$) | 82.12 | 39.82 | 50.90 | 76.51 |
> | BERT ($\tau = 0.4$) | 82.25 | **47.49** | 52.34 | 77.18 |
> | BERT ($\tau = 0.6$) | **83.86** | 43.16 | **59.21** | **77.56** |
> | BERT ($\tau = 0.8$) | 83.85 | 43.16 | 52.70 | 77.18 |
>
> | Model | STSB (Spear. $\uparrow$)  | CoLA (MCC $\uparrow$) | RTE (ACC $\uparrow$) | Linear Probing (CoLA, ACC $\uparrow)$ |
> | :---: | :---: | :---: | :---: | :---: |
> | Mistral ($\tau = 0.0$) | 80.34 | 49.81 | 50.90 | 80.06 |
> | Mistral ($\tau = 0.2$) | **80.48** | 49.45 | 50.90 | 81.20 |
> | Mistral ($\tau = 0.4$) | 80.40 | 50.54 | 52.34 | 80.44 |
> | Mistral ($\tau = 0.6$) | 80.29 | **54.15** | **59.21** | **81.40** |
> | Mistral ($\tau = 0.8$) | 80.26 | 52.70 | 52.70 | 80.82 |
>
> #### References
>
> Barbero et al., Locality Aware Graph Rewiring in GNNs, ICLR 2024
>
> Arnaiz-Rodriguez et al., DiffWire: Inductive Graph Rewiring via the Lovász Bound, LoG 2022
>
> Qian et al., Probabilistic Graph Rewiring via Virtual Nodes, NeurIPS 2024
>
> Warstadt et al., Neural Network Acceptability Judgments. TACL 2019
>
> Bowman et al., A Fast Unified Model for Parsing and Sentence Understanding, ACL 2016
>
> Hewitt & Manning (2019)., A Structural Probe for Finding Syntax in Word Representations., 2019

---

> > ### Author Response · Authors · 2025-11-21
> > **Response (Part 2)**
> >
> > **Q2 & Q4** How does efficiency, scalability, and computational cost of GLOT compare against different backbones and tasks?
> >
> > **A2 & A4**. Thank you for the important questions. We have grouped them into one response to provide a comprehensive analysis of GLOT’s efficiency across architectures, tasks, and sequence lengths. To address these comments, we conducted two new sets of experiments: *(1)* a cross-model/cross-task efficiency benchmark; and *(2)* a long-context scalability stress test to quantify the $\mathcal{O}(L^2)$ graph construction cost. We discuss these experiments below and report in Appendix D of our revised paper.
> >
> > **1. Cross-Model and Cross-Task Generalization (Addressing W2):** In our submission, we measured efficiency focused with Mistral/CoLA, because Mistral is a large model (7B) that we view as a good test-bed for our efficiency . Nonetheless, and inspired by your comment, we have expanded this study to compare GLOT with Full Fine-Tuning (FT) and LoRA across both Encoder-only (BERT) and Decoder-only (Mistral-7B) backbones on three diverse tasks: CoLA (Grammar), STSB (Similarity), and RTE (Entailment). The runtimes are measured on an NVIDIA A6000 GPU with a batch size of 32.
> >
> > The results are reported in the Table below.As can be seen from the Table, our GLOT consistently achieves $\approx 100\times$ speedups and massive memory reductions regardless of the backbone architecture, validating that the efficiency gains are intrinsic to the method and not specific to one model. These results are added in Table 9 of Appendix D.2 in our revision. Thank you.
> >
> > | Method   | No. Trainable Params | GPU Memory (GB) $\downarrow$ | Batch Runtime (ms) $\downarrow$ | MCC $\uparrow$ | Spear. $\uparrow$ | ACC $\uparrow$ |
> > | :--- | :---: | :---: | :---: | :---: | :---: | :---: |
> > | **Mistral-7B (Decoder)** | | | | | | |
> > |   Full FT + [EOS]   |   7.11B | 32.59 | 1318.8 $\pm$ 1.1 | 49.63 | 55.68 | 55.23 |
> > | LoRA (r = 64) + [EOS] | 167.8M | 33.50 | 1454.6 $\pm$ 1.1 | 48.23 | 54.54 | 53.43 |
> > | GLOT (ours) | **8.92M** | **0.42** | **13.4 $\pm$ 3.0** | **53.29** | **80.51** | **59.21** |
> > | **BERT-Base (Encoder)** | | | | | | |
> > |   Full FT + [CLS]   |   109.5M | 0.74 | 52.5 $\pm$ 0.1 | 38.31 | 60.10 | 56.68 |
> > | LoRA (r = 64) + [CLS] | 10.7M | 0.86 | 77.4 $\pm$ 0.2 | 36.16 | 64.64 | 56.32 |
> > | GLOT (ours) | **8.92M** | **0.42** | **13.4 $\pm$ 3.0** | **47.49**  | **83.86** | **59.21** |
> >
> >
> >
> >
> >
> >
> >
> >
> >
> >
> > **2. Graph Construction & Long-Context Scalability (Addressing W4): ** Regarding graph construction, we followed your guidance and benchmarked the graph construction step as well as the total forward pass runtime across maximum context lengths ($L_{max}$) of different models, extending up to 32K tokens. The results are reported in the Table below, which we also added to the paper in Table 10 of Appendix D.2.
> >
> > From the Table below, we can see that the graph construction runtime portion compared with the overall runtime of the LLM is relatively small. Notably, even at 32,768 tokens, graph construction takes only $\approx 0.3$ seconds, representing only 1.3% of the total inference time. This confirms that the computational bottleneck remains the LLM backbone itself. Thank you.
> >
> > |  Backbone  | Max Context Length    | Graph Construction (ms) | Total Runtime (ms) | Overhead |
> > |:----------:|:---:|:------------------:|:--------------:|:---------------------:|
> > | BERT / RoBERTa |  512  | 0.043 $\pm$ 0.002  | 5.36 $\pm$ 0.06 | 0.8% |
> > | SmolLM2 | 8192 | 4.455  $\pm$   0.047 | 772.30 $\pm$ 0.11 | 0.5% |
> > | TinyLlama - 1.1 B| 2048 | 0.672  $\pm$   0.001 | 143.15 $\pm$ 0.05 | 0.6% |
> > | Llama - 3B| 8192 | 15.046 $\pm$ 0.373 | 2041.77 $\pm$ 0.19 | 0.7% |
> > | Mistral-7B | 32768 |  303.466 $\pm$   1.024  | 23460.29 $\pm$ 4.77| 1.3% |

---

> > > ### Author Response · Authors · 2025-11-21
> > > **Response (Part 3)**
> > >
> > > **Q3**. How are the contributions of GLOT positioned relative to the related work discussed in Section 2?
> > >
> > > **A3**. We appreciate the opportunity to clarify GLOT's positioning. As discussed in Section 3 (L223-230 in the original submission), our paper explicitly frames prior pooling methods like AdaPool as instances of the “DeepSets” framework where tokens are processed as an independent set.  Nonetheless, we agree with you that connecting this discussion with our theoretical distinction and our empirical evidence is beneficial. Therefore, to accommodate your comment, we revised Section 2 in our paper, and we highlighted that the “relational learning” advantage of GLOT is not just theoretical, but directly validates our synthetic experiment in Table 7. GLOT maintains $>97%$ accuracy even in the presence of 90% noisy tokens, while the performance of set-based pooling methods struggle filtering noisy tokens. Additionally, in Section 2 in our original submission, we discussed how GLOT is distinct from prior graph-based NLP works (e.g., TextGCN, Yao et al., 2019) that typically rely on global corpus-level statistics or fixed syntactic dependency trees. To address your comment, we elaborate on this discussion in the revised paper:  GLOT introduces a latent graph construction mechanism that builds semantic token graphs based entirely on the hidden space of the LLM. This allows GLOT to recover rich structural information specific to the current context without requiring external parsers or expensive fine-tuning, establishing it as a novel and lightweight paradigm for adapting frozen language models. We added this discussion to our related work in Section 2. Thank you.
> > >
> > > ---
> > >
> > > ### Summary
> > >
> > > We appreciate your constructive feedback that helped us to improve the quality of the paper, in our opinion. We remain committed to addressing any additional questions you may have, and hope that you will consider our responses in your final assessment.  Thank you.

---

### Author Response · Authors · 2025-11-21

We thank the reviewers for their thoughtful and constructive feedback. We are encouraged that the reviewers recognized the **novelty** of reframing pooling as relational learning (**gK63**, **BcGt**), the **practical value** of operating on frozen backbones (**56SJ**, **aPBo**) and the **robustness** of GLOT to signal dilution (**56SJ**, **aPBo**). We are also pleased that **Reviewer 56SJ** found our method "highly practical" for resource-constrained deployments and **Reviewer gK63** noted the methodology is presented "clearly and intuitively."

In response to your constructive and thoughtful feedback, we have revised our paper to strengthen its positioning and empirical and validation. Below is a summary of the major updates and additional experiments included in our revision:

1. **Expanded MTEB Evaluation:** To address the coverage concerns raised by  Reviewers **gk63** and **aPBo**, we significantly expanded our evaluation on the MTEB benchmark using the Mistral-7B backbone. The results further support the strong performance of GLOT. We see that in many cases GLOT achieves competitive or better performance than state-of-the-art techniques with frozen-backbone baselines on tasks requiring explicit dependency modeling and signal preservation, significantly outperforming Mean Pooling on **Summarization** (+41.2% relative improvement), **Retrieval** (+18.3%), and **Pair Classification** (+8.2%).

2. **Comparison to prompting-based sentence embedding methods:** Following Reviewer **gK63**’s suggestion, we compared GLOT against prompting-based methods (PromptBERT, PromptEOL, Pretrained CoT and Knowledge Enhancement). GLOT consistently outperforms these parameter-free approaches, for example, by +18.8% on BERT and +5.0% on Mistral-7B on the STS-B task. These results further confirm that learning from token interactions is more effective than current prompt based methods for our tasks.

3. **Comparison to contrastive fine-tuning based sentence embedding methods:** Addressing Reviewer **aPBo**, we benchmarked GLOT against fully fine-tuned contrastive models (SBERT, SimCSE). GLOT achieves superior performance on complex GLUE tasks (e.g., +22.5 points on CoLA vs. SBERT) and competitive performance on semantic similarity, all while using ~92x fewer parameters and a frozen backbone.

4. **Benchmarking the cost of graph construction:** We profiled the exact cost of the graph construction step (Reviewers **56SJ** and **aPBo**). We found that this step is relatively small, taking only ~0.04ms for standard lengths and constituting **less than 1.3% of total inference latency even at 32K context length**. Furthermore, we extended our efficiency analysis to include encoder-only models (BERT), confirming that GLOT delivers **$>100\times$** speedups across different architectures.

5. **Theoretical clarifications and additional ablations:** We clarify that the performance improvement of GLOT is attributed to the fact that standard pooling methods for frozen LLM backbones treat tokens as an independent set of vectors, analogous to the DeepSets framework. GLOT, on the other hand, treats tokens as semantic graph in the latent embedding space of the frozen LLM, and  learning a pooled representation from token interactions leads to robust sentence embeddings (Reviewer **BcGt**). Additionally, we attribute the performance of GLOT to the token-graph by ablating against a parameter-matched deeper MLP (Reviewer **aPBo**). Overall, with a take-home message that the **paradigm shift of token pooling as relational learning task** leads to efficient adaptation of billion-scale LLMs and superior performance.

We have incorporated these results into the revised manuscript, with all changes marked in blue. We believe these additions comprehensively address your comments, and we feel that your constructive feedback helped us to improve the quality of the paper. We hope that you find our responses satisfactory, and that you will consider revising your scores.

---

### Author Response · Authors · 2025-12-03
**Concluding Comments by Authors (Part 1)**

Dear Area Chair,

In light of the conclusion of the discussion period that ended earlier than expected, we would like to provide a concise summary of the reviews, the rebuttal, and the revisions included in the current version of our submission. We hope this information will be helpful for your decision, and we thank you for your commitment to the reviewing process and attention to the details below.

We would like to use this opportunity to also express our sincere gratitude  to all of our four Reviewers for their detailed and constructive comments that in our opinion were beneficial for improving the quality of our paper. We have implemented your suggestions and follow-up discussions in the revised version, with changes marked in blue. Moreover, During the rebuttal period, we systematically addressed every point raised by each reviewer (evaluation scope, baselines, and theoretical rationale) and incorporated all clarifications and significant new experiments into the revised paper. We believe that our clarifications and revisions fully address your comments. Thank you.

---

## **1. Overall assessment and scores after rebuttal before the discussion freeze**
The ratings and status are as follows:

**Reviewer 56SJ**

- Soundness: 2 (fair) -> Addressed via new ablations.
- Presentation: 3 (good)
- Contribution: 3 (good)
- **Rating: 6**

**Reviewer gK63**

- Soundness: 3 (good)
- Presentation: 3 (good)
- Contribution: 3 (good)
- **Rating: 6**

**Reviewer aPBo**

- Soundness: 2 (fair) -> Addressed via extended MTEB evaluation and efficiency profiling.
- Presentation: 2 (fair)
- Contribution: 2 (fair)
- **Rating: 4->6**
- **Status:** The Reviewers participated in the discussion, and  in their final comment **raised the score to 6**, and expressing their satisfaction our rebuttal and responses. We have also responded to the last comment that included insightful suggestions, which we added to the revised paper.

**Reviewer BcGt**

- Soundness: 3 (good)
- Presentation: 3 (good)
- Contribution: 2 (fair)
- **Rating: 4**
- **Status:** We provided the requested theoretical clarification (set vs. graph-based pooling) and ablations with various GNN architectures. **The Reviewer could not participate in the discussion because of the early discussion termination**. Nonetheless we find that our responses address all of their comments.

Therefore our scores prior to the discussion freeze are **6, 6, 6, 4**, with the Reviewer assigning 4 (BcGt) not participating in the discussion, for which we provided full and detailed responses, which we believe could result in a score revision.

---


## **2. Strengths highlighted by the Reviewers**

**Conceptual novelty and paradigm shift:** Reviewers recognized the value of reframing pooling as relational learning over token graphs. **Reviewer 56SJ** highlighted "Reframing sentence pooling as token-relation modeling" as a core strength. **Reviewer BcGt** noted the "clear motivation" and that the method is "sound."

**Efficiency Analysis:** There was strong consensus on the method's utility. **Reviewer 56SJ** called it "highly practical" with "high efficiency." **Reviewer gK63** noted the "significant practical value" of operating on frozen LLMs. **Reviewer aPBo** appreciated the "lightweight" architecture (approx. 9M parameters).

**Robustness:** Reviewers were impressed by the method's stable performance in the presence of distractors. **Reviewer 56SJ** specifically cited the "strong robustness to noise" demonstrated in our diagnostic stress test (97% accuracy with 90% distractors).

---

> ### Author Response · Authors · 2025-12-03
> **Concluding Comments by Authors (Part 2)**
>
> ## **3. Main concerns and how they were resolved**
>
> **Evaluation Scope: Expanded MTEB Evaluation and New Baselines (gK63, aPBo)**
>
> Concern: Reviewers requested broader evaluation on MTEB and comparisons to prompting/contrastive fine-tuning methods.
>
> Resolution: We significantly expanded the MTEB evaluation to **over 250 new experiments** (Table 12, 50 tasks, 5 methods), where GLOT outperforms Mean Pooling by large margins (+41.2% on Summarization). We added comparisons to **PromptBERT**, **PromptEOL** (Table 13), and **SBERT/SimCSE** (Table 14), showing GLOT outperforms these parameter-free and fine-tuned baselines. **Reviewer aPBo** acknowledged this resolution in their final comment.
>
> **Theoretical Rationale: Why do GNNs improve the performance significantly? (BcGt, 56SJ)**
>
> Concern: Queries regarding the specific choice of GNN and the influence of graph construction.
>
> Resolution: We clarified the theoretical distinction between "set-based" pooling (DeepSets) and "graph-based" pooling (GLOT). We added ablations comparing GLOT to **GCN**, **GIN**, **and a parameter-matched MLP** (Tables 11, 15), proving that the performance gains stem from relational learning, not just added parameters.
>
> **Computational Cost and Scalability (56SJ, aPBo)**
>
> Concern: Questions about the $O(L^2)$ graph construction cost and cross-model efficiency.
>
> Resolution: We profiled the graph construction (Table 10), showing it consumes **<1.3%** of batch run time even at 32K context length. We provided a unified efficiency comparison (Table 17) demonstrating GLOT is **~100x faster** than LoRA/Fine-tuning while achieving superior performance.
>
> ---
>
> ## **4. Effect of the discussion freeze**
>
> The reverted scores do not fully reflect the extent of the revisions, and we hope that these important details will be taken into consideration in the final decision making process:
> - **Reviewer aPBo (Score 6 after discussion)** engaged late in the process to express satisfaction with our commitment to the full MTEB suite. Their concerns about are now concretely resolved by the expanded tables in the revised paper. They also **raised the score to 6**.
> - **Reviewer BcGt (Score 4)** asked for a rationale for the GNN's success. We provided both a theoretical basis (DeepSets vs. Graphs) and empirical proof (GLOT vs. MLP ablation). As they rated "Soundness" as Good, we believe these clarifications resolve their primary reservation regarding the method's justification.
> - Beyond that, not all Reviewers had the chance to communicate with us, and although their initial rating was already positive, we believe that our clarifications and added experiments requested by them could have led to further score increases.
>
> ---
>
> ## **5. Conclusion**
>
> In summary, prior to the discussion freeze, we have three positive ratings (6) and one borderline ratings (4) where the reviewers' specific concerns have been addressed through extensive new experiments (MTEB evaluation, efficiency analysis, new baselines) and the Reviewer could not participate in the discussion because of the OpenReview incident. Overall, our GLOT presents a novel, efficient, and highly robust paradigm for sentence representation that bridges the gap between frozen LLMs and fine-tuning.
>
> We kindly ask from you, our Area Chair, to base your decision on this complete rebuttal, discussion, and revised submission record, which we believe is thoroughly address all Reviewers’ comments, and that you will consider the overall positive evaluation and feedback received from our Reviewers.
>
>
> We are grateful for your commitment to the reviewing process and for your attention to these important details.
>
> Thank you, and kindest regards, \
> Authors

---

### Meta-Review · Area_Chair_FZxH · 2026-01-20

**Summary:**

Reviewers’ pre-rebuttal concerns focused on whether the method is sufficiently justified and rigorously evaluated.
The main issues were: (1) the rationale/theory for why adding a GNN over a latent token graph yields large gains (and what part of the design is essential), (2) evaluation scope and baselines, especially that the MTEB coverage was limited and comparisons to stronger / newer baselines (e.g., prompting-based sentence embedding methods and contrastive fine-tuning methods like SBERT/SimCSE) were missing, (3) attribution and ablations, including requests for parameter-matched alternatives (e.g., an MLP pooling head) and different GNN variants, and (4) efficiency/scalability validation, since efficiency claims were initially reported on a narrow setting and did not fully account for graph construction and long-context behavior.

**Reviewer Concerns:**

The authors addressed the reviewers' concerns as follows:

Reviewer 56SJ: The authors added deeper analysis of the token-graph construction (e.g., topology/threshold sensitivity) and strengthened the efficiency/scalability evidence by reporting more complete profiling, explicitly including the overhead of graph construction and longer-context settings.

Reviewer gK63: The authors expanded the evaluation to a much larger portion of MTEB, added comparisons to stronger prompting-based embedding baselines, and clarified applicability on decoder-only backbones

Reviewer aPBo: The authors added missing strong baselines (contrastive/fine-tuned sentence embedding methods), included parameter-matched ablations (e.g., MLP pooling head vs GNN) and multiple GNN variants to isolate what drives gains, and broadened the efficiency analysis to be more comprehensive and comparable.

Reviewer BcGt: The authors strengthened the theoretical/technical justification by adding a clearer framing of why relational/message-passing modeling can outperform set-based pooling, and supported it with the new ablations (GNN variants and parameter-matched alternatives) to show improvements are not simply due to extra capacity.

**Reviewer Scores:**

Reviewer aPBo: 4 → predicted 6. With the added baselines/ablations/profiling, likely moves to borderline accept

Reviewer BcGt: 4 → predicted 6. Added theory + ablations directly target their core concern; with full discussion they could plausibly move upward, but may remain cautious.

---

### Decision · Program_Chairs · 2026-01-26

Accept (Poster)